# PHAT: Modeling Period Heterogeneity for Multivariate Time Series Forecasting

**Jiaming Ma[1], Qihe Huang[1], Haofeng Ma[3], Guanjun Wang[1], Sheng Huang[1],**
**Zhengyang Zhou[1,2], Pengkun Wang[1,2], Xu Wang[1,2], Binwu Wang[1,2,\*], Yang Wang[1,2,\*]**

[1]University of Science and Technology of China (USTC), Hefei, Anhui, China
[2]Suzhou Institute for Advanced Research, USTC, Suzhou, Jiangsu, China
[3]The University of Nottingham Ningbo China, Ningbo, Zhejiang, China
`JiamingMa@mail.ustc.edu.cn`

## Abstract

While existing multivariate time series forecasting models have advanced significantly in modeling periodicity, they largely neglect the periodic heterogeneity common in real-world data, where variates exhibit distinct and dynamically changing periods. To effectively capture this periodic heterogeneity, we propose PHAT (Period Heterogeneity-Aware Transformer). Specifically, PHAT arranges multivariate inputs into a three-dimensional "periodic bucket" tensor, where the dimensions correspond to variate group characteristics with similar periodicity, time steps aligned by phase, and offsets within the period. By restricting interactions within buckets and masking cross-bucket connections, PHAT effectively avoids interference from inconsistent periods. We also propose a positive-negative attention mechanism, which captures periodic dependencies from two perspectives: periodic alignment and periodic deviation. Additionally, the periodic alignment attention scores are decomposed into positive and negative components, with a modulation term encoding periodic priors. This modulation constrains the attention mechanism to more faithfully reflect the underlying periodic trends. A mathematical explanation is provided to support this property. We evaluate PHAT comprehensively on **14** real-world datasets against **18** baselines, and the results show that it significantly outperforms existing methods, achieving highly competitive forecasting performance. Our sources is available at [GitHub](GitHub).

## 1 Introduction

Multivariate Time Series (MTS) forecasting serves as a core enabling technology for critical applications such as energy demand prediction, traffic management, financial modeling, and healthcare monitoring (Qiu et al., 2024b; Ma et al., 2025b; Huang et al., 2025a; Yue et al., 2025; Li et al., 2026).

Periodicity, as a crucial intrinsic characteristic of time series data, plays a decisive role in enhancing forecasting performance through accurate modeling (Lin et al., 2024a; Zhou et al., 2022b; Lin et al., 2024b). To this end, researchers have developed a series of cutting-edge methods. One line of work modifies neural network architectures to adapt to periodicity (Luo & Wang, 2024). For instance, some studies leverage the strong capability of the Transformer architecture to model long-range dependencies (Zhou et al., 2021; 2022b; Nie et al., 2022). In addition, seasonal-trend decomposition techniques have been widely employed (Hu et al., 2025b;b; Ma et al., 2025d). These methods separate the original time series into seasonal and trend components, which are then modeled by parallel sub-networks. This strategy enables more efficient extraction and utilization of periodic information. Furthermore, recent studies have incorporated classical signal processing tools, particularly frequency domain analysis methods like the Fast Fourier Transform (FFT), to more precisely identify and model periodic patterns in time series data (Ye et al., 2024; Zhang et al., 2025).

---

\*Corresponding authors: Yang Wang and Binwu Wang.

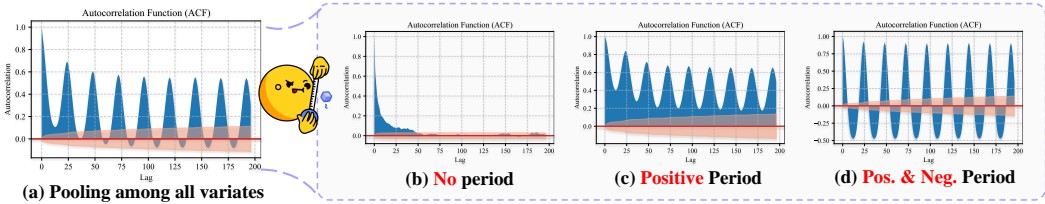

Figure 1: The visualization of period heterogeneity phenomenon on ZafNoo dataset. The orange region in autocorrelation function represents the 95% confidence interval for the null hypothesis from Bartlett's Test (Arsham & Lovric, 2011). Correlation coefficients that fall within this interval are not statistically significant and cannot be rejected as noise (Chandler, 1987).

Despite the substantial progress, two major limitations remain: ❶ Most existing models unify periodic modeling by treating variates as interchangeable channels for pooling and fusion, implicitly assuming a shared, static periodic length. This overlooks the pronounced heterogeneity in periodic behavior across variates. As shown in Figure 1, three variates from the ZafNoo dataset exhibit distinct period lengths. Forcing such diverse periodicities into the unified framework can lead models to learn spurious temporal dynamics. ❷ Mainstream Transformer architectures amplify positive correlations but suppress negative ones during attention normalization, overlooking inverse or complementary dynamics inherent in periodic signals, as shown in Figure 1 (d). Yet such negative correlations offer critical insights into system dynamic. Therefore, it is crucial to develop multivariate time series forecasting models capable of accurately capturing periodic heterogeneity.

In this paper, we propose a $\underline{P}$eriod $\underline{H}$eterogeneity-$\underline{A}$ware $\underline{T}$ransformer (PHAT) for MTS forecasting. Specifically, PHAT restructures multivariate time series into a periodic bucket structure by grouping variates based on their periodic lengths. Within each bucket, each sequence is further reshaped into a 2D tensor—rows align time steps by phase, while columns capture offsets within the period. Interactions are restricted to variates within the same bucket, enabling the model to capture diverse periodic patterns, while cross-bucket links are masked to prevent interference between variates with differing periodicities. Second, PHAT introduces a Positive-Negative Self-Attention mechanism (PNA), which interprets periodic dependencies through two attention coefficients: phase alignment and periodic offset. The periodic offset coefficient is further decomposed into positive and negative components, with a modulation term encoding periodic priors. This term reduces the positive weights and increases the negative weights for distant phase-aligned points (and vice versa), enabling the attention mechanism to more faithfully capture the periodic structure.

**Contributions**. ❶ *Initial Exploration*. We relax the single-period assumption to handle complex time series with heterogeneous periodicities and introduce PHAT, the first method expressly developed to model such period heterogeneity. ❷ *Periodic Bucket*. PHAT introduces a "periodic bucket" structure to manage time series data with heterogeneous periodicity, facilitating the learning of periodic patterns. ❸ *Novel Self-attention*. We further propose a positive-negative attention mechanism that represents periodic information with distinct positive and negative components, combined with a modulation term encoding periodic priors. Its favorable properties are demonstrated mathematically. ❹ *Empirical Validation*. On **14** real-world datasets with **18** baselines, PHAT achieves SOTA performance on approximately **73.95% (71/96)** of the metrics while maintaining low computational complexity. Additionally, PHAT demonstrates strong robustness in complex periodic scenarios.

## 2 RELATED WORK

Periodicity plays a pivotal role in the predictability of time series data (Lin et al., 2025; Wang et al., 2024a; Ma et al., 2025d). Recent advancements have introduced a range of sophisticated techniques aimed at improving the ability to capture long-term dependencies, thereby enhancing the perception of periodic patterns. For example, classical seasonal–trend decomposition uses moving-average kernels for sliding aggregation to extract trend components (Zeng et al., 2023; Kingma & Ba, 2014), while CycleNet proposes a learnable Cycle Decomposition to capture periodicity (Lin et al., 2024a). Architecturally, Transformer-based models excel at modeling long-range dependencies: Autoformer connects sequences by periodically aggregating similar subsequences (Wu et al., 2021),

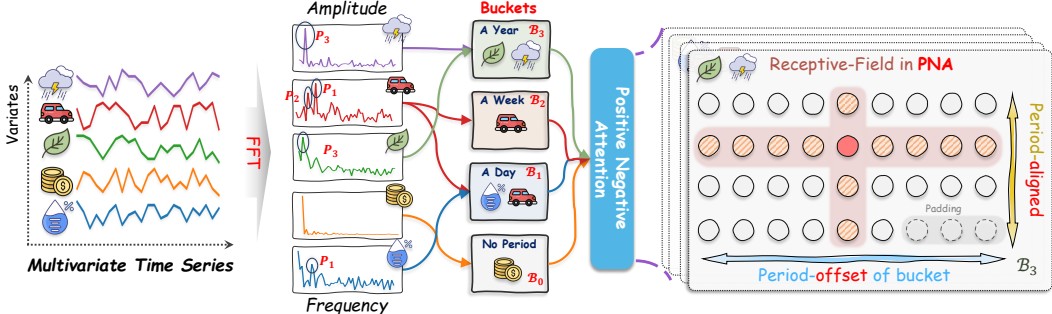

Figure 2: The overall architecture of PHAT. Each bucket contains the same periodic variates and uses PNA to capture the periodic attention mechanism.

while ModernTCN leverages very large convolutional kernels to substantially enlarge the receptive field and capture long-range temporal dependencies (Luo & Wang, 2024). Frequency-domain analysis and multi-scale modeling have also been used to strengthen periodic representations (Zhou et al., 2022b; Wang et al., 2024a).

However, existing methods typically handle multivariate series via pooling or adaptive fusion and model periodicity from a unified perspective, implicitly assuming all variates share the same periodic patterns. This overlooks widespread periodic heterogeneity and often ignores negatively correlated periodic components, which can provide complementary perspectives and valuable information gains. We provide a detailed discussion of multivariate time series forecasting in **Appendix** A.

## 3 METHODOLOGY

Multivariate time-series data typically refer to either a collection of multiple temporal objects or a single object observed through multiple feature channels (Huang et al., 2024b; Wu et al., 2020; Ma et al., 2025e). Given a multivariate time series input $\mathbf{X} = [\mathbf{x}_1, \mathbf{x}_2, \ldots, \mathbf{x}_T] \in \mathbb{R}^{C \times T}$ observed over the past $T$ time steps, where $\mathbf{x}_t \in \mathbb{R}^C$ represents the observation at time step $t$ across $C$ variates. And we use $\mathbf{X}_i \in \mathbb{R}^T$ to represent the input sequence corresponding to variates $i$. The objective of the multivariate time-series forecasting is to forecast the subsequent $L$ time steps $\mathbf{Y} = [\mathbf{x}_{T+1}, \mathbf{x}_{T+2}, \ldots, \mathbf{x}_{T+L}] \in \mathbb{R}^{C \times L}$.

As shown in Figure 2, we propose PHAT for modeling periodic heterogeneity, a common yet previously overlooked property of real-world multivariate time series, thus motivating our PHAT. PHAT includes a novel period bucket structure to manage time series. Subsequently, PHAT integrates a positive-negative attention mechanism to precisely model the positive and negative correlations of periodicity. Finally, we perform weighted fusion of the generated periodic components based on their frequency saliency to produce the prediction results.

### 3.1 PERIOD BUCKET FOR TIME SERIES

#### 3.1.1 PERIOD DETECTION

Fast Fourier Transform (FFT) (Duhamel & Vetterli, 1990) is a commonly used tool for analyzing the characteristics of time series, especially periodicity (Liu, 2025; Wu et al., 2022b; Fang et al., 2023). Therefore, we first apply the FFT to each variate in the input sequence $\mathbf{X}$. Subsequently, we retain only the spectral magnitudes corresponding to the Top-$K$ significant frequency components, which are then converted into discrete period lengths. This process can be expressed as follows:

$$\mathbf{P} = \left\lfloor \frac{T}{\arg \text{Top}_K[|\text{FFT}(\mathbf{X})|]} + 0.5 \right\rfloor \in \mathbb{N}^{K \times C}, \tag{1}$$

where $\arg \text{Top}_K[\cdot]$ selects the indices of the $K$ most salient frequencies. $\mathbf{P}$ represents the set of period lengths from $C$ variates.

### 3.1.2 TIME SERIES PERIOD BUCKET CONSTRUCTION AND REPRESENTATION

To improve periodicity modeling, we propose a novel period-bucket structure for reconstructing time series, consisting of two stages: bucketing and folding.

**Bucketing** This process groups variates by their dominant period lengths. Given $K * C$ periods, we first remove duplicate values, assuming there are $\mathcal{N}$ distinct elements. For any period length $P_i$, we create a bucket matched to it, denoted as $\mathcal{B}_i$, and place the variates with a period length of $P_i$ into the bucket. Because a variate can exhibit multiple periodicities, buckets are not necessarily disjoint. In addition, we assign Bucket-0 to manage variates that do not exhibit significant periodic behavior.

**Folding** This process further reshapes the sequence of each variate. For the input sequence of variate $j$ in Bucket-$b$, denoted as $\mathbf{X}_j$, we first use an linear layer to align it with the future window: $\mathbf{X}_j \to \mathbf{X}_j \in \mathbb{R}^{L \times d}$. It is important to emphasize that the period length of variates in Bucket-$b$ is $P_b$. Next, we segment $\mathbf{X}_j$ into small fragments of length $\lfloor L/P_b \rfloor$, which may require zero-padding for $\mathbf{X}_j$. This process can be expressed mathematically as:

$$\overline{\mathbf{X}}_j = \text{Unflatten}(\mathbf{X}_j^{\text{pad}}; P_b) \in \mathbb{R}^{P_b \times N_b},$$

$$\mathbf{X}_j^{\text{pad}} = \begin{cases} \text{Concat}(\mathbf{X}_j^\top, \mathbf{0}), & \text{if } L \bmod P_b > 0, \\ \mathbf{X}_j^\top, & \text{if } L \bmod P_b = 0. \end{cases} \in \mathbb{R}^{(P_b * N_b)} \quad (2)$$

where $N_b = \lfloor L/P_b \rfloor$ represents the number of periods contained in the sequence. So for Bucket-$b$ that contains of $|\mathcal{B}_b|$ time series, we can generate a 3D bucket-structured input denoted as $\bar{\mathbf{X}}^{(b)} \in \mathbb{R}^{|\mathcal{B}_b| \times P_b \times N_b}$, which adapts to the characteristics of periodic features: ❶ Periodic homogeneous variates are grouped in the first dimension; ❷ Time steps in the second dimension are periodic-offset within the same periodic; ❸ Each time step in the last dimension is periodic-aligned.

Then, we perform interactions between variates to learn their dependencies as follows,

$$\mathbf{Z}^{(b)} = \left(\bar{\mathbf{X}}^{(b)}\right)^\top \mathbf{W}_i + \mathbf{b}_i \in \mathbb{R}^{P_b \times N_b \times d_h}, \quad (3)$$

where $\mathbf{W} \in \mathbb{R}^{|\mathcal{B}_b| \times d_h}$ and $\mathbf{b} \in \mathbb{R}^{d_h}$ are learnable parameters with the dimension size $d_h$. The processing of Bucket with periodicity $\mathcal{B}_0$ is represented in **Appendix** B.

### 3.2 POSITIVE-NEGATIVE X-SHAPE ATTENTION FOR PERIODICITY MODELING

Conventional self-attention allows unrestricted token interactions, which is suboptimal for capturing the periodic structures inherent in time series. To address this, we propose Positive-Negative Attention for Periodicity Modeling (PNA), featuring three key designs: ❶ **X-Shaped Receptive Field**: PNA allocates attention along the rows and columns of the bucketed representation Z , forming an X-shaped (cross-like) receptive field centered on each target element. This structure explicitly separates period-aligned (across-period) and period-offset (within-period) relationships, enhancing periodic pattern modeling. ❷ **Strong Periodic Inductive Bias**: Attention scores are modulated by a periodic-distance-dependent term, enforcing inductive bias that aligns attention weights with the underlying periodic trends. ❸ **Decoupled Positive–Negative Correlation Modeling**: PNA separately model positive and negative periodic dependencies, mitigating interference between opposing correlations and enabling more expressive representation of complex periodic interactions.

Specifically, given Bucket-$b$ representation $\mathbf{Z}^{(b)} \in \mathbb{R}^{P_b \times N_b \times d_h}$ (except for Bucket-0, which lacks periodic characteristics), we first obtain multiple components using the following formula:

$$[\mathbf{Q}_1; \mathbf{Q}_2] = \mathbf{Z}^{(b)} \mathbf{W}_q, \quad [\mathbf{K}_1; \mathbf{K}_2] = \mathbf{Z}^{(b)} \mathbf{W}_k, \quad \mathbf{V} = \mathbf{Z}^{(b)} \mathbf{W}_v, \quad \mathbf{\Lambda} = \sigma(\mathbf{Z}^{(b)} \mathbf{W}_g), \quad (4)$$

where $\mathbf{W}_Q, \mathbf{W}_K, \mathbf{W}_V \in \mathbb{R}^{d_h \times d}, \mathbf{W}_\lambda^h \in \mathbb{R}^{d \times 1}$ are learnable projection parameters. $\mathbf{Q}_1$ and $\mathbf{Q}_2 \in \mathbb{R}^{P_b \times N_b \times 2d}$ are the query vectors, $\mathbf{K}_1$ and $\mathbf{K}_2 \in \mathbb{R}^{P_b \times N_b \times 2d}$ are the key vectors, and $\mathbf{V} \in \mathbb{R}^{P_b \times N_b \times 2d}$ is the value vector. $\mathbf{\Lambda}$ is the weighted strength filter. $\sigma(\cdot)$ is the sigmoid function. The calculation process of PNA can be written as:

$$\text{PNA}(\mathbf{Z}^{(b)}) = \text{Attention}([\mathbf{Q}_1; \mathbf{Q}_2], [\mathbf{K}_1; \mathbf{K}_2], \mathbf{V}, \mathbf{\Lambda}) = \overline{\mathbf{A}} \times_1 \left(\widetilde{\mathbf{A}} \times_2 \mathbf{V}\right), \quad (5)$$

where $\times_i$ means the multiplication in the $i$-th dimension of the matrix. Periodic-aligned attention $\widetilde{\mathbf{A}}$ encodes dependencies between time-steps that share the same phase across periods, while periodic-offset attention $\overline{\mathbf{A}}$ captures dependencies among time-steps within the same period.

**Period-offset Attention $\overline{\mathbf{A}}$**   We compute the positive logits $\boldsymbol{\zeta}$ and negative logits $\boldsymbol{\eta}$ of the period-offset attention with separate query and key in each head as follows,

$$\boldsymbol{\zeta} = \mu \mathbf{Q}_1 \times_1 \mathbf{K}_1^\top \in \mathbb{R}^{P_b \times P_b \times N_b}, \quad \boldsymbol{\eta} = \mu \mathbf{Q}_2 \times_1 \mathbf{K}_2^\top \in \mathbb{R}^{P_b \times P_b \times N_b}, \tag{6}$$

where $\mu = d^{-1/2}$ is the scale factor. The positive and negative logits are first adjusted by the periodic modulation terms and then separately normalized via softmax to produce attention coefficient matrices. These two matrices are subsequently fused to form the final attention matrix as follows,

$$\overline{\mathbf{A}} = \mathrm{Softmax}(\tilde{\boldsymbol{\zeta}}) - \boldsymbol{\Lambda} \odot \mathrm{Softmax}(\tilde{\boldsymbol{\eta}}) \in \mathbb{R}^{P_b \times P_b \times N_b}, \tag{7}$$

$$\tilde{\boldsymbol{\zeta}}[m,n] = \boldsymbol{\zeta}[m,n] - \underbrace{\sum_{s \in \Delta_{m,n}^{(b)}} \mathrm{Softplus}(\boldsymbol{\zeta}[m,s])}_{\text{Positive Modulation Term}}, \quad \tilde{\boldsymbol{\eta}}[m,n] = \boldsymbol{\eta}[m,n] - \underbrace{\sum_{s \in \nabla_{m,n}^{(b)}} \mathrm{Softplus}(\boldsymbol{\eta}[m,s])}_{\text{Negative Modulation Term}},$$

where $\zeta[m,n]$ denotes the attention coefficient between the $m$-th and $n$-th time steps within a signal of period $P_b$, corresponding to the $(m,n)$-th entry of the attention matrix $\zeta$. We define the periodic relative distance between $m$-th and $n$-th time steps as $\delta_{m,n}^b$. For a fixed $m$ and $n$, let $\Delta_{m,n}^p = \{s | \delta_{m,s}^b < \delta_{m,n}^b\} \cup \{m\}$ be the set of time steps whose periodic distance to $m$ is smaller than that between $m$ and $n$. We aggregate the attention coefficients $\zeta[m,s]$ for all $s \in \Delta_{m,n}^p$. After applying the $\mathrm{Softplus}(\cdot)$ activation (Zheng et al., 2015), this term encourages the model to produce attention weights that decay monotonically with increasing periodic distance, thereby reinforcing the inductive bias toward local periodic structure. The periodic distance is computed as follows:

$$\delta_{ij}^b = \min\{(i-j) \bmod \mathcal{B}_b, (j-i) \bmod \mathcal{B}_b\} \in [0, \lfloor \mathcal{B}_b/2 \rfloor], \tag{8}$$

Conversely, the farther apart the periodic positions of the time steps, the greater their negative correlation. Therefore, the set $\nabla_{m,n}^b = \{s | \delta_{m,s}^b > \delta_{m,n}^b\} \cup \{m\}$ includes time steps with larger periodic relative distances. Then, the negative modulation term is computed as the sum of the attention coefficients of these time steps and applied to the attention coefficients.

In this manner, PNA ensures that as the periodic relative distance between two time steps increases, their positive correlation coefficient decreases while their negative correlation coefficient increases. **We provide a mathematical explanation of this favorable property of the periodic-offset attention, which can be found in Appendix C.2.**

The row sums of the generated $\overline{\mathbf{A}}$ are not strictly equal to 1, satisfying:

$$\sum_{j=1}^{\mathcal{B}_b} \overline{\mathbf{A}}[i,j,n] = 1 - \boldsymbol{\Lambda} < 1, \qquad \forall i = \{1, \ldots, \mathcal{B}_b\}; \; n \in \{1, \ldots, N_b\}. \tag{9}$$

To address it, when generating the final attention, we inject a residual path with residual strength $\boldsymbol{\Lambda}^h$ to stabilize the information flow. Please refer to Equation 11.

**Period-aligned Attention $\tilde{\mathbf{A}}$**   To capture dependencies among time steps that are phase-aligned with the periodicity, we employ a simplified self-attention mechanism. This mechanism shares the same query, key, and value vectors as the positive component of the period-offset attention, as these phase-aligned time steps exhibit strong correlations. This process can be expressed as:

$$\tilde{\mathbf{A}} = \mathrm{Softmax}(\mu \mathbf{Q}_1 \times_2 \mathbf{K}_1^\top) \in \mathbb{R}^{P_b \times N_b \times N_b}. \tag{10}$$

where $\mu$ is the learnable coefficient.

**Multi-head Attention Output**   We introduce a multi-head mechanism to enhance the model's expressiveness. The final computation is as follows:

$$\mathrm{Multi\text{-}Head}(\mathbf{Z}^{(b)}) = \mathrm{Concat}(\mathrm{head}^1, \mathrm{head}^2, \ldots, \mathrm{head}^H)\mathbf{W}_O, \tag{11}$$

$$\mathrm{head}^h = \boldsymbol{\gamma}_h \mathrm{Tanh}[\alpha_h(\mathrm{PNA}(\mathbf{Z}^{(b)}) + \boldsymbol{\Lambda}^h \odot \mathbf{Z}^{(b)})] + \boldsymbol{\beta}_h, \tag{12}$$

where $H$ is the number of heads, $\mathbf{W}_O \in \mathbb{R}^{d \times d}$ is learnable parameters, and $\alpha_h \in \mathbb{R}, \boldsymbol{\gamma}_h, \boldsymbol{\beta}_h \in \mathbb{R}^{2*d_h}$ are learnable normalization parameters. $\odot$ means Hadamard Product. We use Dynamic Tanh (Zhu et al., 2025) to eliminate any remaining numerical instability.

A special case involves the variates in the zero bucket, which lack periodicity. For these variates, the folding operation is skipped. When computing period-aligned attention and period-offset attention, absolute distance is used instead of periodic distance. The details are represented in **Appendix** B.

### 3.3 BUCKET-WISE FORECASTING

**Flatten&Align**  Let us define the output of the $b$-th bucket after PNA learning as $\bar{\mathbf{Z}}^{(b)} \in \mathbb{R}^{P_b \times N_b \times d}$. Then, we flatten its first two dimensions, and if zero-padding has been applied (i.e., $(P_b * N_b) > L$), we truncate the padded time steps: $\mathbb{R}^{P_b \times N_b \times d} \to \mathbb{R}^{L \times d}$. Finally, we align the channels to the number of variates originally in the bucket:

$$\widetilde{\mathbf{Z}}^{(b)} = \bar{\mathbf{Z}}^{(b)} \mathbf{W}_i^{(b)} + \mathbf{b}^{(b)} \in \mathbb{R}^{|\mathcal{B}_b| \times L} \tag{13}$$

where $\mathbf{W}^{(b)} \in \mathbb{R}^{d \times |\mathcal{B}_b|}$ and $\mathbf{b}^{(b)} \in \mathbb{R}^{|\mathcal{B}_b|}$ are learnable parameters. $|\mathcal{B}_b|$ denotes the number of variates in the $b$-th bucket.

**Frequency-based Multi-period Prediction**  To predict the future value of the $c$-th variate among $C$ variates, we first determine which buckets contain this variate based on its $K_c$ period lengths, and then extract the corresponding bucket representations for prediction, which can be expressed as:

$$\hat{\mathbf{Y}}_c = \sum_{b=1}^{K_c} \alpha_c^{(b)} \tilde{\mathbf{Z}}_c^{(b)} \in \mathbb{R}^L, \qquad \text{s.t.} \quad c \in \{\mathcal{B}_i\}_{i=1}^{\mathcal{N}}, \quad (b = 1, \dots, \mathcal{N}) \tag{14}$$

$$\alpha_c^{(b)} = \text{Avg}(\text{Softmax}(|\beta_c^{(b)}|)), \quad \beta_c^{(b)} = \text{Extract}\left(\text{FFT}(\mathbf{X}_c^{(b)})\right) \tag{15}$$

where $\beta_c^{(b)}$ denotes the spectral magnitude extracted in the frequency domain for period length $b$. After normalization, $\beta_c^{(b)}$ is converted into the weight $\alpha_c^{(b)}$, which is used to perform a weighted fusion of the corresponding bucket representation $\tilde{\mathbf{Z}}_c^{(b)}$, yielding the final prediction. Finally, we can generate the prediction for $C$ variates $\hat{\mathbf{Y}} \in \mathbb{R}^{C \times L}$.

## 4 EXPERIMENTS

### 4.1 EXPERIMENTAL SETUP

**Protocol Settings**  All experiments are performed on an NVIDIA A100 GPU with 80 GB of memory using PyTorch. For a comprehensive evaluation, we disable the "Drop Last" batch sampling procedure (Li et al., 2024; Qiu et al., 2024a). Optimization is performed with Adam (Kingma & Ba, 2014). Mean squared error (MSE) and mean absolute error (MAE) are used for evaluation. Considering that different models vary in their sensitivity to input history, we treat the look back length $T$ as a tunable hyperparameter and report the best performance, and also comparisons using a fixed input length in Section D.3. The details of hyperparameters are summarized in Table 4.

Table 1: Statistics of used datasets.

| Datasets | NN5 | Exchange | FRED-MD | ETTh | ETTm | AQShunyi | AQWan | ILI | CzeLan | ZafNoo | NASDAQ | NYSE |
|---|---|---|---|---|---|---|---|---|---|---|---|---|
| # Samples | 791 | 7,588 | 728 | 14,440 | 57,600 | 35,064 | 35,064 | 966 | 19,934 | 19,225 | 1,244 | 1,243 |
| # Frequency | 1 day | 1 day | 1 month | 1 hour | 15 mins | 1 hour | 1 hour | 1 week | 30 mins | 30 mins | 1 day | 1 day |
| Split Ration | 7:1:2 | 7:1:2 | 7:1:2 | 6:2:2 | 6:2:2 | 6:2:2 | 6:2:2 | 7:1:2 | 7:1:2 | 7:1:2 | 7:1:2 | 7:1:2 |
| Damain | Banking | Economic | Economic | Electricity | Electricity | Environment | Environment | Health | Nature | Nature | Stock | Stock |

**Datasets**  We evaluate our method on **15** datasets: 14 real-world benchmarks (NN5, Exchange, FRED-MD, ETTh1, ETTh2, ETTm1, ETTm2, AQShunyi, AQWan, ILI, CzeLan, ZafNoo, NASDAQ, NYSE) and one synthetic dataset. Synthetic dataset is constructed by concatenating sequences from ETTm1 (period 96) and ETTh1 (period 24) along the time axis to simulate varying periodicities (see Appendix D.4). To account for varying dataset sizes, we adopt two evaluation regimes: for datasets with fewer than 5,000 samples, the tunable range of look back length is $T \in \{36, 104\}$ and

Table 2: Multivariate time series forecasting performance comparison. We report average MSE and MAE. Best results are **bold** , second-best are underlined .

| Methods | Ours 2025 | | TimeKan 2025 | | xPatch 2025 | | Amplifier 2025 | | CycleNet 2024 | | TimeMixer 2024 | | SparseTSF 2024 | | iTransformer 2024 | | Pathformer 2024 | | PDF 2024 | |
|---|---|---|---|---|---|---|---|---|---|---|---|---|---|---|---|---|---|---|---|---|
| Metrics | MSE | MAE | MSE | MAE | MSE | MAE | MSE | MAE | MSE | MAE | MSE | MAE | MSE | MAE | MSE | MAE | MSE | MAE | MSE | MAE |
| **ETTh** 96 | **0.316** | **0.356** | 0.324 | 0.37 | 0.327 | 0.364 | 0.334 | 0.377 | 0.327 | 0.369 | 0.327 | 0.376 | 0.322 | 0.364 | 0.342 | 0.377 | 0.326 | 0.364 | 0.318 | 0.366 |
| 192 | **0.360** | **0.394** | 0.364 | 0.399 | 0.384 | 0.403 | 0.381 | 0.407 | 0.374 | 0.400 | 0.381 | 0.409 | 0.371 | 0.395 | 0.398 | 0.422 | 0.377 | 0.398 | 0.366 | 0.398 |
| 336 | **0.384** | **0.412** | 0.394 | 0.421 | 0.420 | 0.427 | 0.413 | 0.435 | 0.401 | 0.422 | 0.402 | 0.432 | 0.387 | **0.408** | 0.419 | 0.439 | 0.408 | 0.421 | 0.396 | 0.421 |
| 720 | 0.412 | 0.441 | 0.425 | 0.453 | 0.456 | 0.460 | 0.431 | 0.456 | 0.438 | 0.458 | 0.444 | 0.46 | **0.402** | **0.431** | 0.460 | 0.466 | 0.444 | 0.459 | 0.427 | 0.448 |
| **ETTm** 96 | **0.224** | **0.288** | 0.225 | 0.3 | 0.225 | 0.291 | 0.233 | 0.303 | 0.229 | 0.298 | 0.229 | 0.301 | 0.241 | 0.306 | 0.238 | 0.31 | 0.227 | 0.293 | 0.225 | 0.296 |
| 192 | 0.272 | 0.323 | 0.285 | 0.334 | 0.276 | 0.325 | 0.277 | 0.333 | 0.274 | 0.327 | 0.280 | 0.335 | 0.284 | 0.334 | 0.292 | 0.346 | 0.278 | **0.323** | 0.270 | 0.327 |
| 336 | **0.311** | **0.349** | 0.316 | 0.358 | 0.320 | 0.352 | 0.321 | 0.359 | 0.319 | 0.354 | 0.323 | 0.359 | 0.323 | 0.357 | 0.328 | 0.367 | 0.322 | 0.352 | 0.312 | 0.357 |
| 720 | 0.388 | **0.392** | 0.380 | 0.402 | 0.401 | 0.405 | 0.393 | 0.404 | 0.390 | 0.398 | 0.393 | 0.402 | 0.398 | 0.402 | 0.402 | 0.412 | 0.395 | 0.397 | **0.379** | 0.399 |
| **NN5** 24 | **0.681** | **0.551** | 0.769 | 0.605 | 0.853 | 0.663 | 1.400 | 0.910 | 0.754 | 0.588 | 0.723 | 0.574 | 1.679 | 1.026 | 0.727 | 0.568 | 0.769 | 0.602 | 0.736 | 0.579 |
| 36 | **0.640** | **0.538** | 0.7 | 0.585 | 0.785 | 0.634 | 1.268 | 0.862 | 0.687 | 0.571 | 0.657 | 0.557 | 1.789 | 1.066 | 0.664 | 0.552 | 0.701 | 0.583 | 0.676 | 0.567 |
| 48 | **0.613** | **0.532** | 0.659 | 0.565 | 0.716 | 0.601 | 1.202 | 0.842 | 0.656 | 0.564 | 0.630 | 0.55 | 1.982 | 1.097 | 0.633 | 0.543 | 0.668 | 0.573 | 0.616 | 0.538 |
| 60 | 0.603 | 0.531 | 0.651 | 0.568 | 0.673 | 0.579 | 1.237 | 0.861 | 0.642 | 0.568 | 0.612 | 0.543 | 1.734 | 1.046 | 0.615 | 0.537 | 0.655 | 0.570 | **0.599** | **0.533** |
| **Exchange** 96 | **0.083** | 0.203 | 0.089 | 0.208 | 0.087 | 0.204 | 0.084 | 0.203 | 0.093 | 0.216 | 0.084 | 0.207 | 0.116 | 0.243 | 0.086 | 0.205 | 0.088 | 0.208 | **0.083** | **0.200** |
| 192 | **0.169** | 0.295 | 0.183 | 0.304 | 0.188 | 0.306 | 0.179 | 0.300 | 0.184 | 0.307 | 0.178 | 0.300 | 0.203 | 0.325 | 0.177 | 0.299 | 0.183 | 0.304 | 0.172 | **0.294** |
| 336 | **0.316** | **0.406** | 0.338 | 0.421 | 0.332 | 0.422 | 0.337 | 0.419 | 0.329 | 0.416 | 0.376 | 0.451 | 0.3600 | 0.437 | 0.331 | 0.417 | 0.354 | 0.429 | 0.323 | 0.411 |
| 720 | **0.773** | **0.669** | 0.918 | 0.719 | 0.899 | 0.718 | 0.929 | 0.719 | 0.854 | 0.698 | 0.884 | 0.707 | 0.882 | 0.714 | 0.846 | 0.693 | 0.909 | 0.716 | 0.820 | 0.682 |
| **FRED-MD** 24 | **21.495** | **0.903** | 25.178 | 0.875 | 35.290 | 1.030 | 42.125 | 1.192 | 31.558 | 0.945 | 29.132 | 0.93 | 66.664 | 1.55 | 28.581 | 0.917 | 43.434 | 1.142 | 30.949 | 0.936 |
| 36 | **38.363** | **1.161** | 45.811 | 1.194 | 52.339 | 1.262 | 70.560 | 1.525 | 56.633 | 1.285 | 58.770 | 1.33 | 96.051 | 1.841 | 54.221 | 1.276 | 70.370 | 1.441 | 56.870 | 1.276 |
| 48 | **52.065** | **1.453** | 76.144 | 1.542 | 73.426 | 1.495 | 102.150 | 1.825 | 124.077 | 2.043 | 90.283 | 1.632 | 139.333 | 2.175 | 89.574 | 1.607 | 96.827 | 1.728 | 88.240 | 1.615 |
| 60 | **76.292** | **1.679** | 96.139 | 1.736 | 103.023 | 1.778 | 145.867 | 2.176 | 172.747 | 2.411 | 126.438 | 1.865 | 190.400 | 2.526 | 130.061 | 1.947 | 134.397 | 2.023 | 131.531 | 1.884 |
| **AQShunyi** 96 | 0.665 | **0.468** | 0.651 | 0.478 | 0.663 | 0.469 | 0.667 | 0.489 | 0.653 | 0.484 | 0.654 | 0.483 | 0.696 | 0.509 | 0.650 | 0.479 | 0.667 | 0.472 | **0.647** | 0.481 |
| 192 | 0.707 | **0.488** | 0.689 | 0.497 | 0.708 | 0.492 | 0.702 | 0.505 | 0.699 | 0.502 | 0.700 | 0.498 | 0.719 | 0.521 | 0.693 | 0.498 | 0.707 | 0.491 | **0.690** | 0.499 |
| 336 | 0.737 | **0.502** | 0.712 | 0.512 | 0.741 | 0.506 | 0.718 | 0.511 | 0.719 | 0.514 | 0.715 | 0.510 | 0.728 | 0.525 | 0.713 | 0.510 | 0.732 | 0.503 | **0.708** | 0.512 |
| 720 | 0.795 | 0.529 | 0.763 | 0.531 | 0.790 | 0.532 | 0.765 | 0.537 | 0.774 | 0.540 | **0.756** | 0.534 | 0.784 | 0.552 | 0.766 | 0.537 | 0.783 | 0.515 | 0.765 | 0.537 |
| **AQWan** 96 | 0.746 | **0.452** | 0.749 | 0.467 | 0.752 | 0.455 | 0.762 | 0.478 | 0.758 | 0.473 | **0.744** | 0.468 | 0.796 | 0.497 | 0.747 | 0.470 | 0.761 | 0.458 | 0.747 | 0.468 |
| 192 | 0.808 | **0.475** | 0.791 | 0.488 | 0.803 | 0.479 | 0.806 | 0.493 | 0.804 | 0.492 | 0.804 | 0.488 | 0.820 | 0.509 | 0.787 | 0.486 | 0.801 | 0.478 | 0.794 | 0.489 |
| 336 | 0.824 | **0.487** | 0.815 | 0.497 | 0.833 | 0.490 | 0.821 | 0.500 | 0.826 | 0.503 | **0.813** | 0.500 | 0.837 | 0.514 | 0.814 | 0.497 | 0.821 | 0.488 | 0.817 | 0.500 |
| 720 | 0.907 | 0.517 | 0.888 | 0.527 | 0.906 | 0.522 | 0.894 | 0.530 | 0.878 | 0.522 | 0.878 | 0.532 | 0.907 | 0.543 | 0.889 | 0.529 | 0.888 | **0.506** | 0.885 | 0.527 |
| **ILI** 24 | **1.318** | **0.705** | 2.176 | 0.928 | 2.320 | 1.017 | 2.950 | 1.220 | 2.195 | 1.023 | 1.804 | 0.820 | 4.917 | 1.660 | 1.783 | 0.846 | 2.086 | 0.922 | 1.801 | 0.874 |
| 36 | **1.523** | **0.775** | 2.166 | 0.993 | 2.255 | 1.011 | 2.759 | 1.187 | 1.971 | 0.942 | 1.891 | 0.926 | 4.377 | 1.518 | 1.746 | 0.860 | 1.912 | 0.882 | 1.743 | 0.867 |
| 48 | **1.437** | **0.775** | 2.011 | 0.928 | 2.221 | 1.004 | 2.821 | 1.199 | 1.950 | 0.951 | 1.752 | 0.866 | 4.412 | 1.525 | 1.716 | 0.898 | 1.985 | 0.905 | 1.843 | 0.926 |
| 60 | **1.461** | **0.777** | 2.010 | 0.967 | 2.269 | 1.013 | 2.857 | 1.196 | 1.996 | 0.966 | 1.831 | 0.930 | 4.497 | 1.521 | 2.183 | 0.963 | 1.999 | 0.929 | 1.845 | 0.925 |
| **CzeLan** 96 | **0.170** | 0.204 | 0.197 | 0.248 | 0.173 | 0.205 | 0.174 | 0.223 | 0.178 | 0.229 | 0.175 | 0.230 | 0.205 | 0.256 | 0.177 | 0.239 | 0.172 | 0.213 | 0.177 | 0.232 |
| 192 | 0.206 | **0.227** | 0.203 | 0.257 | 0.196 | 0.228 | 0.204 | 0.251 | 0.209 | 0.252 | 0.206 | 0.254 | 0.223 | 0.284 | 0.201 | 0.257 | 0.207 | 0.236 | 0.205 | 0.252 |
| 336 | 0.236 | **0.254** | 0.239 | 0.291 | 0.218 | 0.256 | 0.232 | 0.274 | 0.242 | 0.280 | 0.230 | 0.277 | 0.261 | 0.308 | 0.232 | 0.282 | 0.240 | 0.262 | 0.232 | 0.277 |
| 720 | 0.285 | **0.293** | 0.248 | 0.308 | 0.258 | 0.296 | 0.254 | 0.325 | 0.282 | 0.315 | 0.262 | 0.309 | 0.330 | 0.362 | 0.261 | 0.311 | 0.288 | 0.298 | 0.273 | 0.310 |
| **ZafNoo** 96 | 0.439 | **0.383** | 0.427 | 0.398 | 0.433 | 0.396 | 0.443 | 0.404 | 0.446 | 0.409 | 0.441 | 0.396 | 0.510 | 0.452 | 0.439 | 0.408 | 0.435 | 0.391 | 0.438 | 0.401 |
| 192 | 0.503 | **0.429** | 0.485 | 0.433 | 0.498 | 0.437 | 0.495 | 0.442 | 0.503 | 0.445 | 0.498 | 0.444 | 0.542 | 0.475 | 0.505 | 0.443 | 0.501 | 0.432 | 0.498 | 0.440 |
| 336 | 0.547 | **0.456** | 0.533 | 0.464 | 0.541 | 0.463 | 0.545 | 0.469 | 0.543 | 0.467 | 0.543 | 0.466 | 0.586 | 0.493 | 0.555 | 0.473 | 0.551 | 0.461 | 0.545 | 0.463 |
| 720 | 0.592 | **0.477** | 0.606 | 0.503 | 0.595 | 0.478 | 0.590 | 0.496 | 0.584 | 0.494 | 0.588 | 0.498 | 0.616 | 0.514 | 0.591 | 0.501 | 0.596 | 0.483 | **0.580** | 0.486 |
| **NASDAQ** 24 | **0.416** | **0.473** | 0.541 | 0.517 | 0.503 | 0.500 | 0.753 | 0.630 | 0.631 | 0.570 | 0.720 | 0.612 | 1.363 | 0.876 | 0.570 | 0.540 | 0.572 | 0.568 | 0.611 | 0.541 |
| 36 | **0.610** | **0.600** | 1.013 | 0.732 | 0.865 | 0.665 | 1.179 | 0.784 | 0.886 | 0.675 | 0.951 | 0.699 | 1.223 | 0.854 | 0.691 | 0.600 | 0.836 | 0.683 | 0.862 | 0.683 |
| 48 | **0.949** | **0.731** | 1.202 | 0.779 | 1.128 | 0.794 | 1.425 | 0.881 | 1.321 | 0.854 | 1.214 | 0.795 | 1.310 | 0.884 | 1.188 | 0.773 | 1.227 | 0.798 | 1.235 | 0.803 |
| 60 | **0.961** | **0.779** | 1.414 | 0.846 | 1.445 | 0.836 | 1.352 | 0.862 | 1.535 | 0.888 | 1.136 | 0.817 | 1.110 | 0.818 | 1.325 | 0.820 | 1.421 | 0.855 | 1.426 | 0.841 |
| **NYSE** 24 | **0.161** | **0.251** | 0.224 | 0.304 | 0.210 | 0.296 | 0.280 | 0.362 | 0.237 | 0.314 | 0.253 | 0.369 | 0.512 | 0.497 | 0.225 | 0.302 | 1.102 | 0.814 | 0.261 | 0.331 |
| 36 | **0.298** | **0.352** | 0.322 | 0.360 | 0.366 | 0.387 | 0.640 | 0.599 | 0.395 | 0.411 | 0.638 | 0.621 | 0.669 | 0.566 | 0.392 | 0.409 | 1.598 | 1.002 | 0.385 | 0.406 |
| 48 | **0.445** | **0.427** | 0.475 | 0.447 | 0.693 | 0.558 | 0.761 | 0.599 | 0.555 | 0.484 | 0.747 | 0.663 | 0.937 | 0.691 | 0.529 | 0.480 | 1.910 | 1.116 | 0.689 | 0.562 |
| 60 | **0.619** | **0.525** | 0.660 | 0.534 | 0.886 | 0.646 | 0.950 | 0.685 | 0.759 | 0.584 | 1.027 | 0.807 | 1.219 | 0.814 | 0.687 | 0.557 | 2.123 | 1.188 | 0.810 | 0.616 |
| **# Top1** | 71 | | 6 | | 2 | | 0 | | 0 | | 4 | | 3 | | 2 | | 3 | | 9 | |
| **# Top2** | 81 | | 23 | | 24 | | 4 | | 1 | | 13 | | 7 | | 12 | | 14 | | 24 | |

forecasting horizons are $L \in \{24, 36, 48, 60\}$; for larger datasets, the tunable range of look back length is $T \in \{96, 336, 512\}$ and forecasting horizons are $L \in \{96, 192, 336, 720\}$. Dataset statistics are summarized in Table 1.

**Baselines** We compare our model with **18** advanced multivariate time series forecasting baselines including TimeKAN (Huang et al., 2025b), xPatch (Stitsyuk & Choi, 2025), Amplifier (Fei et al., 2025), CycleNet (Lin et al., 2024a), TimeMixer (Wang et al., 2024b), SparseTSF (Lin et al., 2024b), iTransformer (Liu et al., 2023), Pathformer (Chen et al., 2024), PDF (Dai et al., 2024), FITS (Xu et al., 2023), PatchTST (Nie et al., 2022), Crossformer (Zhang & Yan, 2023), NLinear (Zeng et al., 2023), TimesNet (Wu et al., 2022a), FEDformer (Zhou et al., 2022b), Triformer (Cirstea et al., 2022), FiLM (Zhou et al., 2022a) and Non-stationary Transformer (Liu et al., 2022b).

## 4.2 FORECASTING PERFORMANCE COMPARISON

Due to space limitations, we averaged the results for ETTh1 and ETTh2 (denoted as ETTh) as well as ETTm1 and ETTm2 (denoted as ETTm). Complete results and additional baseline comparison experiments can be found in **Appendix 7**.

As shown in Table 2, among the advanced baselines, TimeKAN leverages the Kolmogorov-Arnold network to approximate the complex nonlinear dynamics in time series, demonstrating effective expressive power. xPatch introduces exponential seasonal-trend decomposition and amplifies low-energy

components in the spectrum, thereby enhancing sensitivity to subtle signals. Some architectures emphasize periodicity modeling. For example, PDF incorporates a multi-scale patch strategy to jointly capture periodic features, while TimeMixer introduces multi-scale modeling techniques, achieving remarkable performance. In the next section, we provide a detailed comparison of these periodicity-learning models' performance on datasets with periodic heterogeneity. In conclusion, due to its strong adaptability to periodic heterogeneity, PHAT achieves SOTA performance. Notably, on the NYSE dataset, it improves MSE by up to **23.33%**. It achieves the best results on **73.95%** of metrics and ranks in the top two for **84.38%** of metrics in Table 7.

### 4.3 ABLATION STUDIES

We design several ablation variants of PHAT to validate the contribution of its key components. Specifically, "**w/o POA**" removes the period-offset attention branch from PNA, while "**w/o PAA**" eliminates the period-aligned attention branch. "**w/o Attn**" cancels the self-attention mechanism entirely, reducing it to a single feed-forward net. "**w/o Bucket**" disables the period-based variate grouping, treating each variate independently rather than within its detected period bucket.

As shown in Figure 3 (a) (*where taller bars indicate lower MSE for intuitive display*), the "**w/o Bucket**" variant exhibits significantly larger prediction errors, indicating that interactions between variates with different periodic characteristics increase the difficulty of capturing complex periodic patterns. The poor performance of the "**w/o PAA**" variant highlights the importance of cross-period synchronization signals at the same phase, especially on datasets with weaker periodicity, such as NN5 and CzeLan. "**w/o POA**" variant achieves extremely poor predictive performance. We further conducted ablation experiments on the combination of periodic offset attention, and the results are shown in Table 10. We found that modeling dependencies between time steps from both positive and negative perspectives is beneficial. Additionally, the modulators configured for each perspective help generate attention coefficients that align more closely with periodic trends. Detailed comparison of combination ablation study on additional datasets is in Appendix D.8.

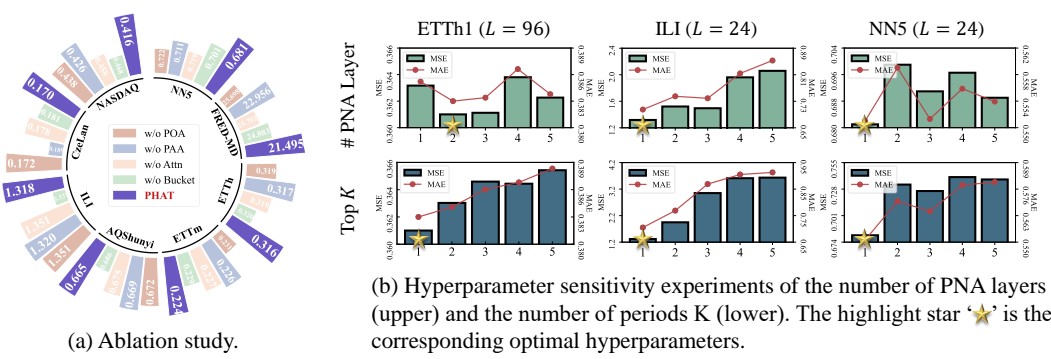

(a) Ablation study.

(b) Hyperparameter sensitivity experiments of the number of PNA layers (upper) and the number of periods K (lower). The highlight star '⭐' is the corresponding optimal hyperparameters.

Figure 3: Ablation study and hyperparameters sensitivity experiments of phats

### 4.4 HYPERPARAMETER SENSITIVITY EXPERIMENTS

We further investigate the sensitivity of two key hyperparameters: the number of PNA layers and the number of cycle lengths $K$ selected for each variate, as shown in Figure 3. Experimental results (Figure 3 (b-upper)) show that our model achieves strong performance with just one or two layers, thanks to the periodic bucket structure and the PNA mechanism, which effectively capture periodic patterns. In contrast, adding more Transformer layers brings little gain and may even hurt performance due to overfitting. As for the hyperparameter $K$, Figure 3(b-lower) shows that on datasets with simple periodicity, a single dominant period ($K$=1) is often sufficient. Using more periods tends to introduce noise and degrade prediction accuracy.

### 4.5 COMPLEXITY ANALYSIS

We evaluate PHAT against advanced baselines on two datasets. As summarized in Table 3, PHAT achieves the best forecasting accuracy while substantially reducing model complexity. Compared with

Transformer-based methods (e.g., PatchTST, PDF) and recently advanced methods (e.g. TimeKAN, xPatch), PHAT cuts parameter counts by more than an order of magnitude and reduces MACs and FLOPs by over 98%. Even compared to lightweight architectures like TimeKAN, PHAT retains a Transformer-style backbone while reducing the parameter count by up to **91.27%** with competitive inference latency. Because the computational cost of our attention scales with the square of the detected period length rather than with the full input sequence, so long sequences remain inexpensive to process. In practice, PHAT also typically requires only a single layer to capture temporal dependencies effectively.

Table 3: Computational Complexity Comparison. # Para: All learnable parameters requiring gradient descent. MACs: multiply–accumulate operations. # FLOPs: floating point operations. K: Kilo ($10^3$). M: Million ($10^6$). B: Billion ($10^9$). Inf: actual inference latency (s).

| | Models | MAE | # Para | # MACs | # FLOPs | Inf lat | | Models | MAE | # Para | # MACs | # FLOPs | Inf lat |
|---|---|---|---|---|---|---|---|---|---|---|---|---|---|
| ETTm1 ($L=96$) | FEDformer | 0.463 | 3.4 M | 1.7 B | 1.3 B | 20.044 s | ETTh1 ($L=720$) | FEDformer | 0.488 | 16.8 M | 111.9 B | 95.7 B | 0.480 s |
| | TimesNet | 0.378 | 2.4 M | 72.2 B | 72.2 B | 3.348 s | | TimesNet | 0.495 | 665.9 K | 77.0 B | 76.9 B | 1.022 s |
| | Crossformer | 0.367 | 2.1 M | 13.5 B | 14.3 B | 1.806 s | | Crossformer | 0.514 | 11.6 M | 60.4 B | 61.5 B | 4.031 s |
| | PDF | 0.340 | 2.0 M | 4.4 B | 4.6 B | 4.234 s | | PDF | 0.462 | 534.2 K | 1.6 B | 1.6 B | 0.351 s |
| | TimeMixer | 0.345 | 397.3 K | 3.2 B | 3.1 B | 3.967 s | | TimeMixer | 0.484 | 2.3 M | 34.0 B | 34.0 B | 0.891 s |
| | TimeKAN | 0.346 | 52.8 K | 205.4 M | 455.0 M | 6.542 s | | TimeKAN | 0.463 | 374.4 K | 1.4 B | 1.7 B | 1.934 s |
| | **PHAT** | **0.330** | **33.4 K** | **2.9 M** | **7.0 M** | **1.671** s | | **PHAT** | **0.458** | **32.7 K** | **3.7 M** | **16.5 M** | 0.462 s |

## 4.6 VISUALIZATION OF PERIODIC POSITIVE-NEGATIVE COMPONENTS IN DATA, FEATURE, AND ATTENTION LEVELS

In this subsection, we further demonstrate that positive-negative periodic correlation indeed exist at all multiple levels. We provide comprehensive experimental analysis at three levels-sequence, feature, and attention—to support our motivation on ETTh1 dataset of variates with daily period and NN5 with weekly period.

❶ **Sequence Level**. We visualize the autocorrelation function (ACF) of the original time series data **X**, which demonstrates the presence of both positive and negative correlations between time steps.

❷ **Feature Level**. We computed the autocorrelation function for each time step in the bucket representation $\mathbf{Z}^{(b)}$. As shown in Figure 4 (a), across each hidden feature dimension, the resulting high-dimensional features continue to exhibit negative correlation, indicating that the positive and negative correlations present in the original signals persist after the complex-valued linear projection.

❸ **Attention Level**. We aim to demonstrate two points: (1) attention coefficients do contain negative components; and (2) the softmax operation tends to smooth out or suppress these negative components. As shown in Figure 4 (b) and (c), we visualized PHAT model's attention logits before and after the softmax, and compared them with a control variant in which our positive–negative attention is replaced by standard self-attention. The results show that before softmax, both models' logits exhibit clear positive and negative components; after softmax, the negative components of the standard self-attention nearly vanish, whereas our attention preserves negative correlations.

In contrast, our PNA mechanism preserves these negative correlations by separately modulating the positive and negative pathways, which is beneficial for capturing the complete periodic structure.

## 4.7 ATTENTION VISUALIZATION

As shown in Figure 5, we illustrate the autocorrelation coefficients between different time steps within a time series (left side) for various datasets, while the right side visualizes the corresponding cycle-shifted attention weights generated by our method. The attention matrix exhibits a clear periodic structure: bright bands represent positive attention, dark bands represent negative attention, and green areas indicate zero attention. We can observe that the trend between our generated attention matrix and the periodic correlation of time series data is consistent, demonstrating the ability to capture

underlying periodic structures. For time series with weaker negative periodicity (such as ZafNoo), there are fewer negative dark regions in the attention. For non-periodic time series (such as those from the ETTh2 and NASDAQ datasets), our cycle-shifted attention mechanism naturally degrades to focus on each individual time step. Overall, our method effectively captures the periodic dynamics of time series.

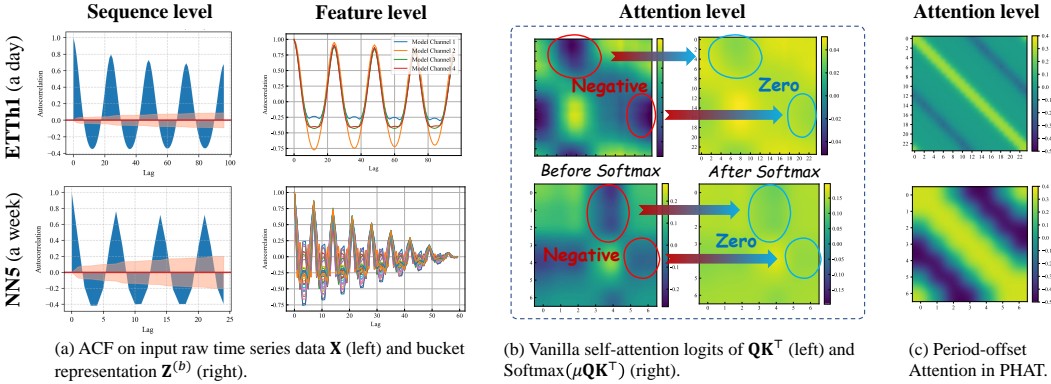

(a) ACF on input raw time series data $\mathbf{X}$ (left) and bucket representation $\mathbf{Z}^{(b)}$ (right).

(b) Vanilla self-attention logits of $\mathbf{QK}^{\top}$ (left) and Softmax($\mu\mathbf{QK}^{\top}$) (right).

(c) Period-offset Attention in PHAT.

Figure 4: Multi-level visualizations on raw-sequence level, feature level and attention level.

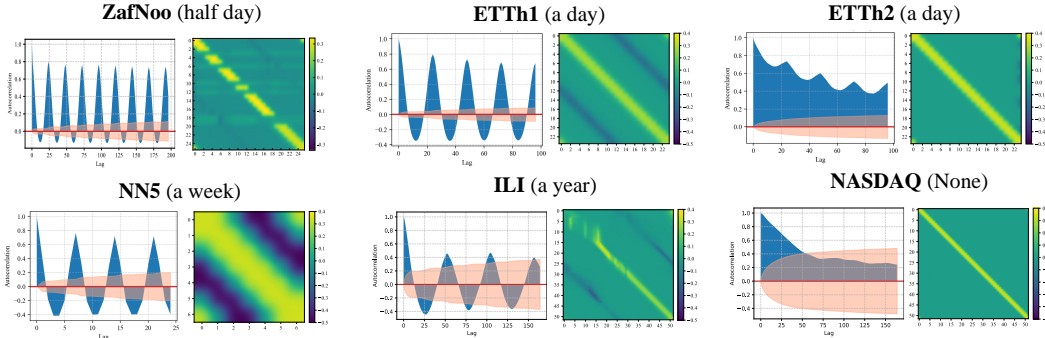

Figure 5: Autocorrelation function (left) and periodic-offset attention weights (right). Highlighted regions denote positive components; dark regions denote negative components.

## 5 CONCLUSION

In this paper, we identify the challenge of periodic heterogeneity, where the periodic characteristics of variates within a dataset differ. To address this, we propose a Periodic Heterogeneity-Aware Transformer (PHAT). PHAT employs a periodic bucket structure to manage multivariate time series with heterogeneous periodicity. By introducing the periodic bucket mechanism and the novel Positive-Negative Attention (PNA) module, PHAT is able to seamlessly and accurately model heterogeneous periodic patterns while capturing both positive and negative phase dependencies. Extensive experiments conducted on 14 real-world datasets demonstrate that PHAT achieves state-of-the-art performance across most evaluation metrics.

## ETHICS STATEMENT

Our work only focuses on the time series forecasting problem, so there is no potential ethical risk.

## REPRODUCIBILITY STATEMENT

In the main text, we have strictly formalized the model architecture with equations. All the implementation details are included in the Appendix, including dataset descriptions, metrics, model, and experiment configurations. The code is provided at an anonymous link.

## ACKNOWLEDGMENT

This paper is partially supported by the National Natural Science Foundation of China (No.12227901) and the Natural Science Foundation of Jiangsu Province (BK20250482). The AI-driven experiments, simulations and model training were performed on the robotic AI-Scientist platform of Chinese Academy of Science.

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

CONTENTS

## A  RELATED WORK

**Multivariate Time Series Forecasting** Deep learning has become the dominant paradigm for multivariate time series (MTS) forecasting thanks to its ability to model nonlinear dynamics and long-range dependencies (Wu et al., 2025c;a; Ma et al., 2025f;c; Huang et al., 2026; Hu et al., 2026). Researchers have developed numerous specialized and general-purpose methods for time series modeling architectures (Qiu et al., 2024b; Ma et al., 2026b; Wang et al., 2026a), channel interaction methodologies (Liu et al., 2026a; Hu et al., 2025c; Qiu et al., 2025c), optimization techniques (Qiu et al., 2025b; Ma et al., 2025e; Wang et al., 2025c;b;a; 2026b;b), and inference strategies (Christou et al., 2024; Ma et al., 2026a). Early solutions relied on recurrent architectures (e.g., LSTM, GRU) or convolutional models such as TCN (Hewage et al., 2020). More recently, the Transformer has emerged as a popular backbone, offering flexible global dependency modeling for sequence-to-sequence tasks and coming to dominate the field (Wen et al., 2022). For example, ModernTCN (Luo & Wang, 2024) uses large convolutional kernels to substantially widen the receptive field and capture broader temporal dependencies, while Pyraformer (Liu et al., 2022a) combines TCN layers with a Transformer backbone to model both local and global patterns. These methods are applied in other complex scenarios within time series domain (Liu et al., 2026b; Qiu et al., 2025a; Wu et al., 2025b; Hu et al., 2025a; Wu et al., 2026).

At the same time, lightweight MLP-based models have gained traction by achieving strong forecasting performance with lower computational cost. DLinear (Zeng et al., 2023) decomposes series into trend and seasonal components via a moving-average kernel and models each with separate linear modules; PatchMLP (Tang & Zhang, 2025) introduces a patch-based approach and channel mixing to enhance inter-variate communication; and HDMixer (Huang et al., 2024a) employs flexible patch lengths to capture short-term dependencies within patches and long-term dependencies across patches while better modeling cross-variate interactions. Overall, Transformer and lightweight MLP approaches present complementary trade-offs between representational capacity and computational efficiency.

**Transformers for Time Series**. Although the Transformer architecture has shown significant promise, it poses challenges such as high computational complexity (Zhang & Yan, 2023; Ma et al., 2025g). To mitigate this, various approaches have been proposed to modify the self-attention mechanism, incorporating techniques like distillation and temporal decomposition to reduce computational demands (Zhou et al., 2021; 2022b; Ma et al., 2025a). LogTrans (Li et al., 2019) introducing a convolutional LogSparse attention that ensured long-distance interactions while reducing the number of interactions, is an early and influential attempt to apply Transformer. Informer (Zhou et al., 2021) proposes ProbSparse attention while combining with a distillation mechanism to select the most representative query vectors to compute the attention scores. More recently, researchers have integrated signal processing methods with Transformers, leveraging techniques such as signal frequency analysis and decomposition (Piao et al., 2024; Zeng et al., 2023). These advancements have enabled models to more effectively capture and analyze the structural characteristics of time series data. However, the softmax operation in Transformers filters out negatively correlated signals—preserving only positive correlations—which leads to a loss of information.

## B  THE BUCKET WITHOUT PERIODICITY

Not all variates in a multivariate time series exhibit clear periodic behavior. To account for such cases, we introduce a dedicated bucket $\mathcal{B}_0$, which collects all variates whose Fourier spectra do not reveal statistically significant periodicity. Although $\mathcal{B}_0$ does not correspond to any explicit cycle, it is still processed through the proposed PNA to maintain architectural consistency and allow residual correlations to be captured. **For notational convenience, we continue to use the terms *period-offset* and *period-aligned* to denote the two orthogonal attention directions, even though no true period exists.** Concretely, learning for bucket $\mathcal{B}_0$ differs from periodic buckets in two key aspects:

❶ **No segmentation into periodic fragments**. Since $\mathcal{B}_0$ has no detected period length, we do not segment its sequences into smaller fragments. Instead, each variate $j$ is withour any padding and folding operation preserving the full temporal structure without periodic folding.

❷ **Offset attention with absolute distance**. In the absence of periodic structure, the offset axis is defined by absolute temporal distance as follows,

$$\delta_{ij}^0 = |j - i| \in [0, L-1],  \tag{16}$$

where the period-offset attention is also coupled with positive attention and negative attention with the same modulation as in periodic buckets. This allows PHAT to still capture local autocorrelations and longer-range and potential temporal recurrences.

An important consequence is that the **period-aligned attention matrix degenerates into the identity** since the period-aligned attention matrix $\tilde{\mathbf{A}} \in \mathbb{R}^{L \times 1 \times 1}$ after softmax normalization along the last axis, all entries are equal to 1. Thus, the PNA attention in $\mathcal{B}_0$ is equal to,

$$\text{PNA}(\mathbf{Z}^{(0)}) = \overline{\mathbf{A}} \times_1 \left( \tilde{\mathbf{A}} \times_2 \mathbf{V} \right) = \overline{\mathbf{A}} \times_1 \mathbf{V} \in \mathbb{R}^{L \times 1 \times d},  \tag{17}$$

Overall, this design ensures that variates without periodicity are handled consistently within the PHAT framework: they bypass unnecessary folding while still benefiting from offset-based attention to capture temporal dependencies. By unifying both periodic and non-periodic buckets under the same framework, PHAT achieves robustness across diverse real-world datasets where not all signals are strictly cyclical.

## C MATHEMATICAL JUSTIFICATION

In this section, we provide mathematical justification to clarify the motivation of PHAT and to address concerns raised during the review process. Section C.1 definite the Period Heterogeneity by mathematical formulation. Section C.2 shows that period-offset attention exhibits periodic behavior aligned with the autocorrelation periodicity of time series. Section C.3 demonstrates that decomposition of positive and negative paths yields strictly lower variance than vanilla attention, offering a clearer learning-theoretic rationale for the effectiveness of our method.

### C.1 A MATHEMATICS DEFINITION OF PERIOD HETEROGENEITY

Here we formally defines period heterogeneity.

**Definition: Period Heterogeneity**. Period heterogeneity refers to the phenomenon where different variates exhibit distinct periodic lengths and periodic correlation patterns.

❶ **Different periodic lengths.** There exist two variates $c_i$ and $c_j$ whose dominant periods differ, i.e.,

$$P_{c_i} \neq P_{c_j},  \tag{18}$$

where $P_{c_i}$ and $P_{c_j}$ denote their corresponding fundamental period lengths.

❷ **Different periodic correlations.** Although periodic patterns are commonly characterized by the presence of correlations across specific time lags, existing models typically overlook the sign of such correlations. In particular, real-world sequences may exhibit negative correlations between certain time steps within a period. Let $\rho_c(\delta) = \text{ACF}(x_{c,t}, x_{c,t+\delta})$ denote the autocorrelation function of variate $c$ at lag $\delta$ within a single period. Then this kind of period heterogeneity exists if there exist a variate $c^*$ such that

$$\text{sign}(\rho_{c^*}(\lfloor P/2 \rfloor)) \ll 0.  \tag{19}$$

This definition reminds us that modeling period heterogeneity not only need to differences in period lengths among variates, but also to the diversity of period correlation patterns—ranging from purely positive correlations to positive and negative correlation patterns.

### C.2 A STATISTICAL INTERPRETATION OF PERIOD-OFFSET ATTENTION (ANOTHER VIEW)

In this subsection, we will show that period-offset attention aligns with the inductive bias of auto-correlation in real-world time series by separating positive and negative paths and assigning them different modulation types. In addition, through a statistical stick-breaking perspective (Ren et al., 2011), we provide an interpretable view of the mechanism and show that the upper bound of the attention logits is monotonic with respect to the period distance.

**Proposition. Period-offset Attention as a Stick-breaking Allocation**     Let $\zeta[m,n]$ and $\eta[m,n]$ denote the raw positive and negative logit between time steps $m$ and $n$ within a detected period of length $P_b$, receptively. Define the modulated positive and negative logits of the period-offset attention as

$$\tilde{\zeta}[m,n] = \zeta[m,n] - \sum_{s \in \Delta_{m,n}^{(b)}} \mathrm{Softplus}(\zeta[m,s]), \tag{20}$$

$$\tilde{\eta}[m,n] = \eta[m,n] - \sum_{s \in \nabla_{m,n}^{(b)}} \mathrm{Softplus}(\eta[m,s]), \tag{21}$$

where $\Delta_{m,n}^{(b)} = \{s \mid \delta_{m,s}^b < \delta_{m,n}^b\} \cup \{n\}$ and $\nabla_{m,n}^{(b)} = \{s \mid \delta_{m,s}^b > \delta_{m,n}^b\} \cup \{n\}$ is the set of offsets closer and farther to $m$ than $n$ under corresponding distance $\delta_{m,s}^b$. Then, before the normalization step in softmax, the exponentiated logits satisfy the following stick-breaking decomposition (Ren et al., 2011):

$$\exp(\tilde{\zeta}[m,n]) = \sigma(\zeta[m,n]) \prod_{\{s \mid \delta_{m,s} < \delta_{m,n}\}} (1 - \sigma(\zeta[m,s])), \tag{22}$$

$$\exp(\tilde{\eta}[m,n]) = \sigma(\eta[m,n]) \prod_{\{s \mid \delta_{m,s} > \delta_{m,n}\}} (1 - \sigma(\eta[m,s])), \tag{23}$$

where $\sigma(\cdot)$ denotes the sigmoid function.

**Proof of Proposition**: The logits from positive branch of period-offset attention is,

$$
\begin{aligned}
&\tilde{\zeta}[m,n] \\
=&\zeta[m,n] - \sum_{s \in \Delta_{m,n}^{(b)}} \mathrm{Softplus}(\zeta[m,s]) \\
=&\zeta[m,n] - \sum_{s \in \Delta_{m,n}^{(b)}} \ln\left(1 + \exp\left(\zeta[m,s]\right)\right) \\
=&\zeta[m,n] - \ln\left(1 + \exp\left(\zeta[m,n]\right)\right) - \sum_{s \in \Delta_{m,n}^{(b)} - \{n\}} \ln\left(1 + \exp\left(\zeta[m,s]\right)\right) \\
=&\ln\left(\frac{\exp\left(\zeta[m,n]\right)}{1 + \exp\left(\zeta[m,n]\right)}\right) + \sum_{s \in \Delta_{m,n}^{(b)} - \{n\}} \ln\left(\frac{1}{1 + \exp\left(\zeta[m,s]\right)}\right) \\
=&\ln\left(\frac{1}{1 + \exp^{-1}\left(\zeta[m,n]\right)}\right) + \sum_{s \in \Delta_{m,n}^{(b)} - \{n\}} \ln\left(1 - \frac{\exp\left(\zeta[m,s]\right)}{1 + \exp\left(\zeta[m,s]\right)}\right) \\
=&\ln\left(\sigma(\zeta[m,n])\right) + \sum_{s \in \Delta_{m,n}^{(b)} - \{n\}} \ln\left(1 - \frac{1}{1 + \exp^{-1}\left(\zeta[m,s]\right)}\right) \\
=&\ln\left(\sigma(\zeta[m,n])\right) + \sum_{s \in \Delta_{m,n}^{(b)} - \{n\}} \ln\left(1 - \sigma(\zeta[m,s])\right),
\end{aligned}
\tag{24}
$$

where $\sigma(\cdot)$ is the sigmoid function and set $\Delta_{m,n}^{(b)} - \{n\} = \{s \mid \delta_{m,s}^b < \delta_{m,n}^b\}$ means all of the closer time step from $m$ than $n$. Hence before the normalization operator in $\mathrm{Softamax}(\cdot)$, the postie logits are activated by exponential function satisfying,

$$
\begin{aligned}
\exp\left(\tilde{\zeta}[m,n]\right) =&\exp\left(\ln\left(\sigma(\zeta[m,n])\right) * \exp\left(\sum_{\{s \mid \delta_{m,s} < \delta_{m,n}\}} \ln\left(1 - \sigma(\zeta[m,s])\right)\right)\right. \\
=&\exp\left(\ln\left(\sigma(\zeta[m,n])\right) * \prod_{\{s \mid \delta_{m,s} < \delta_{m,n}\}} \exp\left(\ln\left(1 - \sigma(\zeta[m,s])\right)\right)\right. \\
=&\sigma(\zeta[m,n]) * \prod_{\{s \mid \delta_{m,s} < \delta_{m,n}\}} (1 - \sigma(\zeta[m,s])).
\end{aligned}
\tag{25}
$$

The proof of the situation of negative logits is similar. □

This identity reveals that the unnormalized attention coefficient assigned to time step $n$ is exactly a stick–breaking allocation. Attention coefficient is allocated sequentially in the order of periodic distance: nearby offsets consume mass first, while farther offsets only receive the leftover. Unlike standard dot-product attention, which distributes weights independently, our period-offset formulation embeds a period-aware stick-breaking bias, yielding structured sparsity, interpretability, and inductive alignment with the autocorrelation structure of time series.

---

**Corollary: Local dominance of periodic distance**

From Proposition 1, it follows that for any reference time step $m$, the allocation weight to time step $n$ in period-offset attention is strictly upper-bounded by the leftover stick mass after all closer offsets have been considered:

$$\exp(\tilde{\boldsymbol{\zeta}}[m,n]) < \prod_{\{s|\delta_{m,s}<\delta_{m,n}\}} (1 - \sigma(\boldsymbol{\zeta}[m,s])), \tag{26}$$

$$\exp(\tilde{\boldsymbol{\eta}}[m,n]) < \prod_{\{s|\delta_{m,s}>\delta_{m,n}\}} (1 - \sigma(\boldsymbol{\eta}[m,s])), \tag{27}$$

---

This corollary formalizes the intuition that local periodic dependencies are always prioritized: nearby (far away) time steps consume most of the positive (negative) attention budget, and more distant steps are suppressed unless strongly supported by their logits. This guarantees that period-offset attention respects the inductive bias of autocorrelation in real-world time series, where closer (farther) periodic positions tend to carry stronger positive (negative) correlations.

## C.3 THE VARIANCE OF PERIOD-OFFSET ATTENTION UNDER PERIOD HETEROGENEITY

As the request from reviewer tCVf, we add a brief learning-theoretic argument to further strengthen the theoretical contribution. Let the vanilla attention between steps $i$ and $j$ be $\mathbf{A}_{ij} = \mathrm{Softmax}(\mu\,\mathbf{Q}_i\mathbf{K}_j^\top)$ with variance $\mathbb{V}[\mathbf{A}_{ij}] = \sigma^2$. Period-offset attention decomposes this interaction into positive and negative paths, leading to

$$\mathbb{V}[\bar{\mathbf{A}}_{ij}] = (1 - \Lambda_{ij})\,\sigma^2, \tag{28}$$

where $\Lambda_{ij} \in (0,1)$ when periodic heterogeneity induces a non-zero negative path. Thus, period-offset attention yields strictly lower variance than vanilla attention. This provides a clear learning-theoretic intuition: the positive and negative decomposition reduces attention-logit variance in heterogeneous periodic settings, producing more stable and accurate estimates than vanilla attention.

## D EXPERIMENTS

### D.1 EXPERIMENTS DETAILS

We report the detailed hyperparameters of our model in Table 4. All experiments were conducted on a single NVIDIA A100 GPU with 80 GB of memory using PyTorch. Forecasting performance was evaluated using mean squared error (MSE) and mean absolute error (MAE). For a comprehensive assessment, we disabled the "Drop Last" batch sampling procedure (Li et al., 2024; Qiu et al., 2024a) and trained models with the Adam optimizer (Kingma & Ba, 2014). Because models vary in sensitivity to input history, the look back length T was treated as a tunable hyperparameter and each model's best performance is reported. Hyperparameter settings are summarized in Table 4.

### D.2 DATASET PERIODICITY TEST

We further visualize temporal correlations across multiple datasets. We use the Autocorrelation Function (Dégerine & Lambert-Lacroix, 2003) to analyze the autoregressiveness between time steps. As shown in Figure 6, most datasets exhibit complex periodic heterogeneity. For example, in the ETTh1 dataset, variate 3 shows only positively correlated periodicity, while variate 4 demonstrates

Table 4: Optimal hyperparameters on our experiments.

| | Hyperparameters | NN5 | Exchange | FRED-MD | ETTh1 | ETTh2 | ETTm1 | ETTm2 | AQShunyi | AQWan | ILI | CzeLan | ZafNoo | NASDAQ | NYSE |
|---|---|---|---|---|---|---|---|---|---|---|---|---|---|---|---|
| L=24/96 | Batch Size | 16 | 16 | 16 | 16 | 32 | 8 | 16 | 32 | 32 | 8 | 16 | 16 | 256 | 32 |
| | Learning Rate | 0.01 | 0.001 | 0.01 | 0.001 | 0.001 | 0.01 | 0.01 | 0.001 | 0.001 | 0.001 | 0.01 | 0.001 | 0.01 | 0.1 |
| | Model Dimension | 32 | 4 | 32 | 4 | 4 | 8 | 4 | 8 | 16 | 24 | 4 | 4 | 24 | 4 |
| | Layers | 1 | 1 | 1 | 2 | 1 | 1 | 1 | 1 | 1 | 1 | 1 | 1 | 1 | 1 |
| | Head Number | 2 | 2 | 2 | 4 | 4 | 4 | 2 | 2 | 2 | 2 | 4 | 2 | 4 | 4 |
| | Look Back Window | 104 | 96 | 36 | 512 | 512 | 336 | 336 | 512 | 512 | 104 | 336 | 336 | 104 | 36 |
| L=36/192 | Batch Size | 256 | 32 | 16 | 16 | 16 | 16 | 16 | 32 | 32 | 64 | 16 | 16 | 16 | 64 |
| | Learning Rate | 0.01 | 0.01 | 0.01 | 0.001 | 0.001 | 0.01 | 0.001 | 0.001 | 0.0001 | 0.01 | 0.0001 | 0.0001 | 0.01 | 0.1 |
| | Model Dimension | 128 | 8 | 96 | 4 | 12 | 8 | 8 | 8 | 8 | 8 | 4 | 6 | 24 | 6 |
| | Layers | 1 | 1 | 1 | 1 | 1 | 1 | 1 | 1 | 1 | 1 | 1 | 1 | 1 | 1 |
| | Head Number | 4 | 4 | 2 | 2 | 2 | 2 | 2 | 2 | 2 | 2 | 4 | 2 | 2 | 2 |
| | Look Back Window | 104 | 96 | 36 | 512 | 512 | 336 | 336 | 336 | 512 | 104 | 512 | 512 | 104 | 36 |
| L=48/336 | Batch Size | 256 | 64 | 16 | 16 | 32 | 16 | 16 | 32 | 32 | 8 | 16 | 16 | 32 | 64 |
| | Learning Rate | 0.01 | 0.01 | 0.01 | 0.001 | 0.01 | 0.0001 | 0.001 | 0.0001 | 0.001 | 0.1 | 0.01 | 0.0001 | 0.1 | 0.1 |
| | Model Dimension | 64 | 8 | 32 | 4 | 16 | 8 | 8 | 16 | 16 | 8 | 4 | 4 | 8 | 8 |
| | Layers | 1 | 1 | 1 | 1 | 1 | 1 | 1 | 1 | 1 | 1 | 1 | 1 | 1 | 1 |
| | Head Number | 4 | 2 | 2 | 2 | 4 | 2 | 2 | 2 | 2 | 2 | 2 | 2 | 2 | 4 |
| | Look Back Window | 104 | 96 | 36 | 512 | 336 | 336 | 336 | 336 | 336 | 104 | 336 | 512 | 104 | 36 |
| L=60/720 | Batch Size | 128 | 256 | 32 | 32 | 128 | 16 | 16 | 32 | 32 | 8 | 16 | 16 | 256 | 64 |
| | Learning Rate | 0.01 | 0.1 | 0.1 | 0.001 | 0.001 | 0.0001 | 0.01 | 0.0001 | 0.0001 | 0.01 | 0.0001 | 0.001 | 0.1 | 0.1 |
| | Model Dimension | 32 | 16 | 32 | 4 | 12 | 4 | 4 | 8 | 8 | 4 | 4 | 4 | 32 | 2 |
| | Layers | 1 | 1 | 1 | 1 | 1 | 1 | 1 | 1 | 1 | 1 | 1 | 1 | 1 | 1 |
| | Head Number | 8 | 2 | 2 | 2 | 4 | 4 | 2 | 4 | 2 | 2 | 2 | 2 | 4 | 2 |
| | Look Back Window | 104 | 96 | 36 | 336 | 336 | 336 | 336 | 336 | 336 | 104 | 512 | 512 | 104 | 36 |

Figure 6: The autocorrelation visualization of ETT datasets.

a bilateral periodic pattern with both positive and negative correlations, and variate 5 exhibits no significant periodicity. In contrast, the ETTh2 dataset lacks clear periodicity across the entire dataset.

## D.3 COMPARISON WITH FIXED INPUT LENGTH SETTINGS

The handling of input windows in time series forecasting tasks has always been flexible. In our experiments, we treated the historical window length as a hyperparameter, allowing models to flexibly select the length based on their own characteristics. We believe that this approach provides a more comprehensive evaluation. In contrast, maintaining a fixed input window length avoids fairness

concerns arising from unequal usage of historical information across models. Here, we do not assess the advantages or disadvantages of these two settings. Instead, we further supplement a comparison under the fixed look back window setting, with the results presented below. As shown in Table 5, Our model still achieves the best performance.

Table 5: Comparison in the same input setting. ETTh1 is 96 to 96 and the others are 24 to 36.

| Dataset | | Ours | TimeKAN | xPatch | Amplifer | CycleNet | TimeMixer | SparseTSF | iTransformer | Pathformer | PDF |
|---|---|---|---|---|---|---|---|---|---|---|---|
| ETTh1 | MSE | **0.375** | 0.378 | 0.379 | 0.443 | 0.400 | 0.381 | 0.400 | 0.386 | 0.386 | 0.378 |
| | MAE | **0.389** | 0.402 | 0.401 | 0.428 | 0.415 | 0.397 | 0.403 | 0.405 | 0.392 | 0.396 |
| ILI | MSE | **2.062** | 2.857 | 2.704 | 3.455 | 2.955 | 2.015 | 5.498 | 2.112 | 2.906 | 2.168 |
| | MAE | **0.876** | 1.053 | 1.064 | 1.296 | 1.195 | 0.863 | 1.777 | 0.885 | 1.154 | 0.899 |
| NASDAQ | MSE | **0.606** | 0.610 | 0.616 | 0.752 | 0.967 | 0.619 | 1.363 | 0.615 | 0.643 | 0.624 |
| | MAE | **0.534** | 0.542 | 0.539 | 0.629 | 0.729 | 0.546 | 0.876 | 0.546 | 0.578 | 0.550 |

## D.4 SYNTHETIC DATASET

We systematically created a synthetic dataset by combining ETTh1 (with a period of 24) and ETTm1 (with a period of 96), which record data from the same variates but at different time granularities. Each dataset was divided into four equal parts along the time axis. Then, we alternately sampled each segment to construct the **Synthetic**, represented as ETTh1[:$\frac{1}{4}$], ETTm1[$\frac{1}{4}$:$\frac{1}{2}$], ETTh1[$\frac{1}{2}$:$\frac{3}{4}$], ETTm1[$\frac{3}{4}$:]. As a result, the synthetic dataset exhibits mixed periodic lengths of (24, 96, 24, 96).

## D.5 DETAILED FORECASTING COMPARISON

We present additional benchmark comparison results in Table 6 and provide fine-grained experimental results for ETTh in Table 7. Among the advanced baseline models, Crossformer achieves strong predictive performance due to its cross-dimension attention mechanism, which facilitates efficient modeling of long-term dependencies. PatchTST, on the other hand, introduces self-supervised learning and adopts a channel-independent strategy to capture complex temporal dynamics. Overall, PHAT achieves state-of-the-art (SOTA) performance, demonstrating exceptional adaptability to periodic heterogeneity. It outperforms other models on the majority of metrics, ranking in the top two positions for 86% of the evaluation metrics, thereby showcasing a significant performance advantage.

## D.6 COMPARISON OF PERIODIC HETEROGENEITY MODELING

We further compared the performance of models specializing in periodic modeling under mixed periodicity scenarios, with results shown in Table 8. ETTh and ETTm exhibit clear periodicity, while ILI and CzeLan show significant differences in periodicity between variates. The NASDAQ dataset displays no periodicity, whereas the Synthetic dataset demonstrates mixed periodicity. Our analysis shows that these periodic learners struggle to handle periodic heterogeneity effectively. On datasets without clear periodic patterns, such as NASDAQ, PDF and SpareTSF exhibit significant performance errors. In contrast, our model remains robust to periodic heterogeneity.

## D.7 COMPARISON OF TRANSFORMER VARIANTS

We further evaluated the effectiveness of the proposed positive-negative attention mechanism (PNA) by comparing it with other attention variants. As shown in Table 9, we found that Crossformer improves representation capability by modeling the dependencies between variates across channels. However, the interaction of variates with different periodic characteristics may introduce confusion. In contrast, the proposed attention mechanism, PNA, achieved the lowest prediction error. This is because it comprehensively models both the positive and negative correlations of periodicity, demonstrating strong periodic representation capabilities.

## D.8 DETAILED ABLATION STUDY

In this section, we conduct combined ablation studies across additional datasets and provide further analysis. Our results show that jointly modeling both positive and negative periodic correlation is

Table 6: Multivariate forecasting results on 14 datasets. We report MSE and MAE. Best results are **bold**, second-best are underlined. The corresponding results on ETTh1 and ETTh2 (denoted ETTh), ETTm1 and ETTm2 (denoted ETTm) are averaged for better presentation.

| Methods | | Ours 2025 | | FITS 2024 | | PatchTST 2023 | | Crossformer 2023 | | NLinear 2023 | | TimesNet 2023 | | FEDformer 2022 | | Triformer 2022 | | FiLM 2022 | | Stationary 2022 | |
|---|---|---|---|---|---|---|---|---|---|---|---|---|---|---|---|---|---|---|---|---|---|
| Metrics | | MSE | MAE | MSE | MAE | MSE | MAE | MSE | MAE | MSE | MAE | MSE | MAE | MSE | MAE | MSE | MAE | MSE | MAE | MSE | MAE |
| ETTh | 96 | **0.316** | **0.356** | 0.327 | 0.371 | 0.326 | 0.367 | 0.570 | 0.519 | 0.331 | 0.371 | 0.362 | 0.391 | 0.358 | 0.400 | 0.668 | 0.543 | 0.326 | 0.370 | 0.469 | 0.456 |
| | 192 | **0.360** | **0.394** | 0.366 | 0.399 | 0.379 | 0.405 | 0.566 | 0.523 | 0.384 | 0.404 | 0.422 | 0.428 | 0.418 | 0.436 | 0.867 | 0.609 | 0.382 | 0.409 | 0.497 | 0.479 |
| | 336 | **0.384** | **0.412** | 0.385 | 0.416 | 0.404 | 0.430 | 0.587 | 0.543 | 0.400 | 0.419 | 0.456 | 0.461 | 0.424 | 0.462 | 0.909 | 0.630 | 0.403 | 0.430 | 0.495 | 0.482 |
| | 720 | 0.412 | 0.441 | **0.409** | 0.442 | 0.432 | 0.459 | 0.945 | 0.698 | 0.423 | 0.447 | 0.478 | 0.472 | 0.479 | 0.488 | 1.025 | 0.690 | 0.444 | 0.4645 | 0.625 | 0.558 |
| ETTm | 96 | **0.224** | **0.288** | 0.234 | 0.300 | 0.227 | 0.299 | 0.305 | 0.379 | 0.232 | 0.298 | 0.265 | 0.322 | 0.340 | 0.386 | 0.313 | 0.366 | 0.233 | 0.299 | 0.313 | 0.352 |
| | 192 | **0.272** | **0.323** | 0.278 | 0.328 | 0.275 | 0.331 | 0.372 | 0.413 | 0.287 | 0.335 | 0.323 | 0.357 | 0.436 | 0.438 | 0.430 | 0.432 | 0.280 | 0.328 | 0.416 | 0.412 |
| | 336 | **0.311** | **0.349** | 0.320 | 0.355 | 0.319 | 0.359 | 0.501 | 0.516 | 0.323 | 0.356 | 0.368 | 0.386 | 0.492 | 0.472 | 0.559 | 0.498 | 0.326 | 0.357 | 0.505 | 0.453 |
| | 720 | **0.388** | **0.392** | 0.390 | 0.397 | 0.389 | 0.402 | 0.752 | 0.613 | 0.396 | 0.400 | 0.444 | 0.428 | 0.536 | 0.501 | 1.209 | 0.666 | 0.393 | 0.400 | 0.595 | 0.506 |
| NN5 | 24 | **0.681** | **0.551** | 0.870 | 0.663 | 0.740 | 0.596 | 0.741 | 0.591 | 0.758 | 0.592 | 0.739 | 0.579 | 0.785 | 0.618 | 1.382 | 0.929 | 0.846 | 0.651 | 1.274 | 0.900 |
| | 36 | **0.640** | **0.538** | 0.814 | 0.655 | 0.694 | 0.595 | 0.703 | 0.589 | 0.693 | 0.577 | 0.717 | 0.585 | 0.727 | 0.606 | 1.352 | 0.920 | 0.883 | 0.702 | 1.318 | 0.93 |
| | 48 | **0.613** | **0.532** | 0.780 | 0.644 | 0.667 | 0.585 | 0.669 | 0.575 | 0.688 | 0.587 | 0.647 | 0.558 | 0.623 | 0.555 | 1.348 | 0.918 | 0.969 | 0.741 | 1.277 | 0.905 |
| | 60 | **0.603** | **0.531** | 0.781 | 0.650 | 0.653 | 0.582 | 0.683 | 0.587 | 0.679 | 0.587 | 0.633 | 0.547 | 0.630 | 0.559 | 1.346 | 0.918 | 0.633 | 0.556 | 1.313 | 0.927 |
| Exchange | 96 | 0.083 | 0.203 | **0.082** | **0.199** | 0.087 | 0.204 | 0.231 | 0.356 | 0.085 | 0.204 | 0.112 | 0.242 | 0.138 | 0.268 | 0.201 | 0.335 | 0.087 | 0.210 | 0.121 | 0.247 |
| | 192 | **0.169** | 0.295 | 0.173 | **0.295** | 0.177 | 0.300 | 0.460 | 0.509 | 0.175 | 0.297 | 0.209 | 0.334 | 0.273 | 0.379 | 0.453 | 0.495 | 0.182 | 0.308 | 0.220 | 0.337 |
| | 336 | 0.316 | 0.406 | 0.317 | 0.406 | **0.297** | **0.399** | 1.034 | 0.825 | 0.320 | 0.409 | 0.358 | 0.435 | 0.437 | 0.485 | 0.703 | 0.630 | 0.318 | 0.409 | 0.352 | 0.437 |
| | 720 | 0.773 | 0.669 | 0.825 | 0.684 | 0.843 | 0.692 | 1.576 | 1.021 | 0.838 | 0.690 | 0.944 | 0.736 | 1.158 | 0.828 | 1.395 | 0.915 | 0.815 | 0.681 | **0.725** | **0.656** |
| FRED-MD | 24 | **21.495** | **0.903** | 56.779 | 1.374 | 32.808 | 0.962 | 385.599 | 3.559 | 32.125 | 0.931 | 43.219 | 1.265 | 66.090 | 1.623 | 395.947 | 5.049 | 40.183 | 1.145 | 47.852 | 1.238 |
| | 36 | **38.363** | **1.161** | 97.396 | 1.774 | 61.035 | 1.345 | 398.728 | 3.716 | 58.332 | 1.258 | 69.554 | 1.531 | 94.359 | 1.863 | 412.165 | 5.210 | 90.434 | 1.670 | 68.140 | 1.493 |
| | 48 | **52.065** | **1.453** | 145.471 | 2.183 | 91.835 | 1.648 | 414.353 | 3.939 | 82.184 | 1.609 | 95.071 | 1.810 | 129.798 | 2.135 | 423.926 | 5.234 | 131.081 | 2.119 | 92.906 | 1.736 |
| | 60 | **76.292** | **1.679** | 196.613 | 2.523 | 127.018 | 1.958 | 422.864 | 4.093 | 109.625 | 1.882 | 116.341 | 1.976 | 173.616 | 2.435 | 432.464 | 5.299 | 180.367 | 2.397 | 117.756 | 1.929 |
| AQShunyi | 96 | 0.665 | **0.468** | 0.655 | 0.485 | **0.646** | 0.478 | 0.652 | 0.484 | 0.653 | 0.486 | 0.658 | 0.488 | 0.706 | 0.525 | 0.665 | 0.492 | 0.664 | 0.486 | 0.771 | 0.518 |
| | 192 | 0.707 | 0.488 | 0.701 | 0.503 | 0.688 | 0.498 | **0.674** | 0.499 | 0.701 | 0.506 | 0.707 | 0.512 | 0.729 | 0.531 | 0.681 | 0.501 | 0.705 | 0.504 | 0.775 | 0.53 |
| | 336 | 0.737 | **0.502** | 0.720 | 0.515 | 0.710 | 0.513 | **0.704** | 0.515 | 0.722 | 0.519 | 0.786 | 0.538 | 0.824 | 0.569 | 0.731 | 0.525 | 0.725 | 0.517 | 0.821 | 0.554 |
| | 720 | 0.795 | 0.529 | 0.774 | 0.540 | 0.768 | 0.539 | 0.747 | **0.518** | 0.777 | 0.545 | 0.756 | 0.528 | 0.794 | 0.561 | **0.742** | 0.534 | 0.782 | 0.544 | 0.793 | 0.547 |
| AQWan | 96 | 0.746 | **0.452** | 0.757 | 0.473 | **0.745** | 0.468 | 0.750 | 0.465 | 0.758 | 0.475 | 0.787 | 0.486 | 0.796 | 0.508 | 0.762 | 0.474 | 0.766 | 0.475 | 0.872 | 0.508 |
| | 192 | 0.808 | 0.475 | 0.806 | 0.492 | 0.793 | 0.490 | **0.762** | 0.479 | 0.809 | 0.496 | 0.778 | 0.489 | 0.825 | 0.517 | 0.786 | 0.484 | 0.809 | 0.494 | 0.860 | 0.524 |
| | 336 | 0.824 | **0.487** | 0.826 | 0.504 | 0.819 | 0.502 | **0.802** | 0.504 | 0.830 | 0.508 | 0.815 | 0.505 | 0.863 | 0.537 | 0.802 | 0.495 | 0.831 | 0.505 | 0.864 | 0.532 |
| | 720 | 0.907 | 0.517 | 0.900 | 0.532 | 0.890 | 0.533 | **0.829** | 0.512 | 0.906 | 0.538 | 0.869 | 0.519 | 0.907 | 0.552 | 0.852 | 0.519 | 0.906 | 0.536 | 0.897 | 0.526 |
| ILI | 24 | **1.318** | **0.705** | 2.182 | 1.002 | 1.932 | 0.872 | 2.981 | 1.096 | 1.998 | 0.919 | 2.131 | 0.958 | 2.398 | 1.02 | 6.052 | 1.730 | 2.256 | 0.996 | 2.394 | 1.066 |
| | 36 | **1.523** | **0.775** | 2.330 | 1.051 | 1.869 | 0.866 | 3.549 | 1.196 | 1.920 | 0.916 | 2.612 | 0.974 | 2.410 | 1.005 | 6.111 | 1.743 | 2.133 | 0.992 | 2.226 | 1.031 |
| | 48 | **1.437** | **0.775** | 2.761 | 1.184 | 1.891 | 0.883 | 3.851 | 1.288 | 1.895 | 0.924 | 1.916 | 0.897 | 2.591 | 1.033 | 6.289 | 1.774 | 2.034 | 0.969 | 2.525 | 1.003 |
| | 60 | **1.461** | **0.777** | 2.929 | 1.217 | 1.914 | 0.896 | 4.692 | 1.450 | 1.964 | 0.947 | 1.995 | 0.905 | 2.539 | 1.070 | 7.000 | 1.893 | 1.974 | 0.929 | 2.410 | 1.010 |
| CzeLan | 96 | 0.178 | **0.204** | 0.187 | 0.243 | 0.176 | 0.232 | 0.581 | 0.443 | 0.178 | 0.228 | 0.179 | 0.239 | 0.231 | 0.311 | 0.818 | 0.519 | 0.180 | 0.232 | 0.240 | 0.296 |
| | 192 | 0.206 | **0.227** | 0.214 | 0.263 | **0.205** | 0.263 | 0.705 | 0.503 | 0.210 | 0.252 | 0.216 | 0.28 | 0.283 | 0.349 | 0.962 | 0.589 | 0.212 | 0.255 | 0.280 | 0.315 |
| | 336 | **0.236** | **0.254** | 0.247 | 0.292 | 0.236 | 0.286 | 0.971 | 0.596 | 0.243 | 0.280 | 0.265 | 0.313 | 0.298 | 0.363 | 1.161 | 0.659 | 0.243 | 0.281 | 0.311 | 0.341 |
| | 720 | 0.285 | **0.293** | 0.291 | 0.329 | 0.270 | 0.316 | 1.566 | 0.762 | 0.290 | 0.326 | 0.283 | 0.338 | 0.426 | 0.449 | 1.496 | 0.741 | 0.282 | 0.312 | 0.328 | 0.379 |
| ZafNoo | 96 | 0.439 | **0.383** | 0.449 | 0.412 | 0.429 | 0.405 | 0.430 | 0.418 | 0.447 | 0.410 | 0.478 | 0.419 | 0.476 | 0.450 | 0.441 | 0.419 | 0.451 | 0.411 | 0.524 | 0.449 |
| | 192 | 0.503 | **0.429** | 0.511 | 0.447 | 0.494 | 0.449 | **0.479** | 0.449 | 0.503 | 0.447 | 0.491 | 0.445 | 0.544 | 0.479 | 0.493 | 0.451 | 0.508 | 0.448 | 0.562 | 0.475 |
| | 336 | 0.547 | **0.456** | 0.544 | 0.468 | 0.538 | 0.475 | **0.505** | 0.464 | 0.545 | 0.470 | 0.551 | 0.480 | 0.628 | 0.523 | 0.523 | 0.467 | 0.549 | 0.471 | 0.614 | 0.502 |
| | 720 | 0.592 | **0.477** | 0.585 | 0.491 | 0.573 | 0.486 | **0.560** | 0.494 | 0.589 | 0.497 | 0.626 | 0.511 | 0.653 | 0.562 | 0.564 | 0.491 | 0.598 | 0.504 | 0.692 | 0.542 |
| NASDAQ | 24 | **0.416** | **0.473** | 0.709 | 0.645 | 0.649 | 0.567 | 1.149 | 0.745 | 0.557 | 0.522 | 0.587 | 0.533 | 0.537 | 0.481 | 2.737 | 1.334 | 0.767 | 0.645 | 0.655 | 0.607 |
| | 36 | **0.610** | **0.600** | 1.058 | 0.778 | 0.821 | 0.682 | 1.414 | 0.885 | 0.869 | 0.668 | 0.792 | 0.664 | 0.808 | 0.628 | 3.387 | 1.534 | 1.379 | 0.835 | 0.991 | 0.695 |
| | 48 | **0.949** | **0.731** | 1.255 | 0.834 | 1.169 | 0.793 | 2.108 | 1.136 | 1.152 | 0.770 | 1.216 | 0.783 | 1.137 | 0.746 | 3.425 | 1.555 | 1.179 | 0.829 | 1.260 | 0.814 |
| | 60 | **0.961** | **0.779** | 1.153 | 0.818 | 1.247 | 0.843 | 2.276 | 1.201 | 1.284 | 0.809 | 1.220 | 0.768 | 1.251 | 0.783 | 3.313 | 1.537 | 1.303 | 0.853 | 1.119 | 0.819 |
| NYSE | 24 | **0.161** | **0.251** | 0.301 | 0.410 | 0.226 | 0.296 | 0.820 | 0.841 | 0.193 | 0.283 | 0.267 | 0.335 | 0.159 | 0.254 | 2.353 | 1.258 | 0.313 | 0.364 | 0.249 | 0.342 |
| | 36 | **0.298** | **0.352** | 0.497 | 0.517 | 0.380 | 0.389 | 0.942 | 0.904 | 0.315 | 0.356 | 0.376 | 0.410 | 0.289 | 0.344 | 3.338 | 1.540 | 0.390 | 0.415 | 0.371 | 0.403 |
| | 48 | **0.445** | **0.427** | 0.741 | 0.633 | 0.575 | 0.492 | 1.049 | 0.955 | 0.464 | 0.438 | 0.573 | 0.506 | 0.477 | 0.457 | 4.248 | 1.733 | 0.538 | 0.48 | 0.521 | 0.468 |
| | 60 | **0.619** | 0.525 | 1.036 | 0.757 | 0.749 | 0.572 | 1.121 | 0.937 | 0.631 | **0.522** | 0.776 | 0.629 | 0.693 | 0.586 | 4.696 | 1.846 | 0.721 | 0.563 | 0.686 | 0.543 |
| # Top1 | | **68** | | 4 | | 8 | | 10 | | 1 | | 1 | | 3 | | 2 | | 0 | | 1 | |
| # Top2 | | **83** | | 14 | | 29 | | 16 | | 20 | | 10 | | 9 | | 7 | | 5 | | 3 | |

beneficial, as it yields a more complete representation of the underlying periodic characteristic. We also observe that using only positive periodicity performs slightly better than using only negative correlations, likely because positive correlations capture the most direct and dominant periodic dynamics. For datasets where negative periodic correlations are weak or largely absent (e.g., ETTh2, ETTm2, and AQShunyi), whether or not negative paths are included—or whether their attenion logits are modulated—has minimal impact on performance. This behavior arises naturally from the gating mechanism $\Lambda$, which adaptively determines the extent to which negative correlations should be modeled based on dataset characteristics. As a result, the model avoids unnecessary overfitting when negative periodicity is not present.

# E DISCUSSIONS

Although both our model and patch-based Transformers rely on operations such as unflattening or folding the sequence, their design philosophies diverge significantly.

Patch-based methods, such as PatchTST (Zeng et al., 2023), Crossformer (Zhang & Yan, 2023), and PDF (Dai et al., 2024), divide the temporal sequence into contiguous segments, reinterpreting one dimension of each patch as a feature axis. This effectively performs temporal down-sampling by

Table 7: Detailed forecasting results on ETTh1, ETTh2, ETTm1 and ETTm2 datasets. We report MSE and MAE. Best results are **bold**, second-best are underlined.

| # Top2 | Methods | | ETTh1 96 | 192 | 336 | 720 | ETTh2 96 | 192 | 336 | 720 | ETTm1 96 | 192 | 336 | 720 | ETTm2 96 | 192 | 336 | 720 |
|---|---|---|---|---|---|---|---|---|---|---|---|---|---|---|---|---|---|---|
| 27 | PHAT (Ours) | MSE | 0.361 | 0.393 | **0.414** | 0.441 | **0.272** | 0.328 | 0.354 | 0.384 | 0.287 | 0.328 | 0.360 | 0.422 | 0.162 | 0.216 | **0.262** | 0.355 |
| | | MAE | **0.383** | 0.410 | 0.427 | 0.458 | **0.330** | 0.379 | 0.398 | 0.425 | **0.330** | 0.362 | **0.381** | **0.412** | 0.246 | 0.285 | 0.317 | **0.372** |
| 5 | TimeKan | MSE | 0.369 | 0.402 | 0.419 | 0.442 | 0.279 | **0.326** | 0.368 | 0.408 | **0.286** | 0.331 | **0.354** | **0.400** | 0.163 | 0.238 | 0.277 | 0.359 |
| | | MAE | 0.396 | 0.417 | 0.430 | 0.463 | 0.343 | 0.380 | 0.411 | 0.443 | 0.346 | 0.368 | 0.386 | 0.417 | 0.254 | 0.299 | 0.330 | 0.387 |
| 8 | xPatch | MSE | 0.378 | 0.420 | 0.467 | 0.516 | 0.275 | 0.348 | 0.373 | 0.395 | 0.288 | 0.331 | 0.367 | 0.447 | 0.162 | 0.202 | 0.272 | 0.354 |
| | | MAE | 0.395 | 0.426 | 0.451 | 0.493 | 0.332 | 0.380 | 0.403 | 0.427 | 0.335 | **0.360** | 0.383 | 0.428 | 0.247 | 0.289 | 0.320 | 0.381 |
| 1 | Amplifier | MSE | 0.373 | 0.414 | 0.442 | 0.455 | 0.295 | 0.348 | 0.383 | 0.407 | 0.292 | 0.327 | 0.365 | 0.427 | 0.174 | 0.226 | 0.276 | 0.358 |
| | | MAE | 0.399 | 0.420 | 0.446 | 0.467 | 0.354 | 0.393 | 0.424 | 0.444 | 0.348 | 0.365 | 0.386 | 0.419 | 0.257 | 0.300 | 0.331 | 0.388 |
| 4 | CycleNet | MSE | 0.374 | 0.406 | 0.431 | 0.450 | 0.279 | 0.342 | 0.371 | 0.426 | 0.299 | 0.334 | 0.368 | 0.417 | **0.159** | **0.214** | 0.269 | 0.363 |
| | | MAE | 0.396 | 0.415 | 0.430 | 0.464 | 0.341 | 0.385 | 0.413 | 0.451 | 0.348 | 0.367 | 0.386 | 0.414 | 0.247 | 0.286 | 0.322 | 0.382 |
| 0 | TimeMixer | MSE | 0.372 | 0.413 | 0.438 | 0.486 | 0.281 | 0.349 | 0.366 | 0.401 | 0.293 | 0.335 | 0.368 | 0.426 | 0.165 | 0.225 | 0.277 | 0.36 |
| | | MAE | 0.401 | 0.430 | 0.450 | 0.484 | 0.351 | 0.387 | 0.413 | 0.436 | 0.345 | 0.372 | 0.386 | 0.417 | 0.256 | 0.298 | 0.332 | 0.387 |
| 10 | SparseTSF | MSE | 0.361 | 0.394 | 0.415 | **0.419** | 0.283 | 0.347 | 0.358 | 0.384 | 0.316 | 0.348 | 0.373 | 0.434 | 0.166 | 0.220 | 0.273 | 0.361 |
| | | MAE | 0.386 | 0.406 | 0.419 | 0.440 | 0.341 | 0.381 | 0.396 | 0.422 | 0.355 | 0.376 | 0.387 | 0.422 | 0.256 | 0.292 | 0.327 | 0.381 |
| 0 | iTransformer | MSE | 0.386 | 0.424 | 0.449 | 0.495 | 0.297 | 0.372 | 0.388 | 0.424 | 0.300 | 0.341 | 0.374 | 0.429 | 0.175 | 0.242 | 0.282 | 0.375 |
| | | MAE | 0.405 | 0.440 | 0.460 | 0.487 | 0.348 | 0.403 | 0.417 | 0.444 | 0.353 | 0.380 | 0.396 | 0.430 | 0.266 | 0.312 | 0.337 | 0.394 |
| 5 | Pathformer | MSE | 0.372 | 0.408 | 0.438 | 0.450 | 0.279 | 0.345 | 0.378 | 0.437 | 0.290 | 0.337 | 0.374 | 0.428 | 0.164 | 0.219 | 0.267 | 0.361 |
| | | MAE | 0.392 | 0.415 | 0.434 | 0.463 | 0.336 | 0.380 | 0.408 | 0.455 | 0.335 | 0.363 | 0.384 | 0.416 | 0.250 | 0.288 | 0.319 | 0.377 |
| 7 | PDF | MSE | **0.360** | **0.392** | 0.418 | 0.456 | 0.276 | 0.339 | 0.374 | 0.398 | **0.286** | **0.321** | **0.354** | 0.408 | 0.163 | 0.219 | 0.269 | **0.349** |
| | | MAE | 0.391 | 0.414 | 0.435 | 0.462 | 0.341 | 0.382 | 0.406 | 0.433 | 0.340 | 0.364 | 0.383 | 0.415 | 0.251 | 0.290 | 0.330 | 0.382 |
| 7 | FITS | MSE | 0.376 | 0.400 | 0.419 | 0.435 | 0.277 | 0.331 | 0.350 | 0.382 | 0.303 | 0.337 | 0.368 | 0.420 | 0.165 | 0.219 | 0.272 | 0.359 |
| | | MAE | 0.396 | 0.418 | 0.435 | 0.458 | 0.345 | 0.379 | 0.396 | 0.425 | 0.345 | 0.365 | 0.384 | 0.413 | 0.254 | 0.291 | 0.326 | 0.381 |
| 1 | PatchTST | MSE | 0.377 | 0.409 | 0.431 | 0.457 | 0.274 | 0.348 | 0.377 | 0.406 | 0.289 | 0.329 | 0.362 | 0.416 | 0.165 | 0.221 | 0.276 | 0.362 |
| | | MAE | 0.397 | 0.425 | 0.444 | 0.477 | 0.337 | 0.384 | 0.416 | 0.441 | 0.343 | 0.368 | 0.390 | 0.423 | 0.255 | 0.293 | 0.327 | 0.381 |
| 0 | Crossformer | MSE | 0.411 | 0.409 | 0.433 | 0.501 | 0.728 | 0.723 | 0.740 | 1.386 | 0.314 | 0.374 | 0.413 | 0.753 | 0.296 | 0.369 | 0.588 | 0.75 |
| | | MAE | 0.435 | 0.438 | 0.457 | 0.514 | 0.603 | 0.607 | 0.628 | 0.882 | 0.367 | 0.410 | 0.432 | 0.613 | 0.391 | 0.416 | 0.600 | 0.612 |
| 1 | NLinear | MSE | 0.385 | 0.422 | 0.431 | 0.439 | 0.276 | 0.345 | 0.368 | 0.406 | 0.301 | 0.355 | 0.372 | 0.430 | 0.163 | 0.218 | 0.273 | 0.361 |
| | | MAE | 0.403 | 0.426 | 0.429 | 0.452 | 0.338 | 0.382 | 0.408 | 0.441 | 0.343 | 0.379 | 0.385 | 0.418 | 0.252 | 0.290 | 0.326 | 0.382 |
| 0 | TimesNet | MSE | 0.389 | 0.440 | 0.523 | 0.521 | 0.334 | 0.404 | 0.389 | 0.434 | 0.340 | 0.392 | 0.423 | 0.475 | 0.189 | 0.254 | 0.313 | 0.413 |
| | | MAE | 0.412 | 0.443 | 0.487 | 0.495 | 0.370 | 0.413 | 0.435 | 0.448 | 0.378 | 0.404 | 0.426 | 0.453 | 0.265 | 0.310 | 0.345 | 0.402 |
| 0 | FEDformer | MSE | 0.379 | 0.420 | 0.458 | 0.474 | 0.337 | 0.415 | 0.389 | 0.483 | 0.463 | 0.575 | 0.618 | 0.612 | 0.216 | 0.297 | 0.366 | 0.459 |
| | | MAE | 0.419 | 0.444 | 0.466 | 0.488 | 0.380 | 0.428 | 0.457 | 0.488 | 0.463 | 0.516 | 0.544 | 0.551 | 0.309 | 0.360 | 0.400 | 0.450 |
| 0 | Triformer | MSE | 0.399 | 0.444 | 0.492 | 0.549 | 0.936 | 1.290 | 1.325 | 1.500 | 0.349 | 0.387 | 0.426 | 0.482 | 0.276 | 0.473 | 0.692 | 1.936 |
| | | MAE | 0.425 | 0.449 | 0.479 | 0.529 | 0.660 | 0.768 | 0.781 | 0.850 | 0.388 | 0.410 | 0.446 | 0.476 | 0.344 | 0.453 | 0.549 | 0.856 |
| 0 | FiLM | MSE | 0.370 | 0.405 | 0.434 | 0.463 | 0.282 | 0.358 | 0.372 | 0.425 | 0.301 | 0.339 | 0.374 | 0.423 | 0.165 | 0.220 | 0.277 | 0.363 |
| | | MAE | 0.394 | 0.416 | 0.435 | 0.474 | 0.346 | 0.401 | 0.425 | 0.455 | 0.343 | 0.365 | 0.385 | 0.414 | 0.254 | 0.291 | 0.329 | 0.386 |
| 0 | Stationary | MSE | 0.591 | 0.615 | 0.632 | 0.828 | 0.347 | 0.379 | 0.358 | 0.422 | 0.415 | 0.494 | 0.577 | 0.636 | 0.21 | 0.338 | 0.432 | 0.554 |
| | | MAE | 0.524 | 0.540 | 0.551 | 0.658 | 0.387 | 0.418 | 0.413 | 0.457 | 0.410 | 0.451 | 0.490 | 0.535 | 0.294 | 0.373 | 0.416 | 0.476 |

Table 8: Comparison with periodicity-modeling model. The results are average MAE among all horizon. $\uparrow$ is the relative percentage increasing regarding PHAT.

| Datasets | ETTh | ETTm | ILI | CzeLan | NASDAQ | Synthetic |
|---|---|---|---|---|---|---|
| TimesNet | $0.438_{\uparrow 9.50\%}$ | $0.373_{\uparrow 10.35\%}$ | $0.933_{\uparrow 23.08\%}$ | $0.292_{\uparrow 19.18\%}$ | $0.687_{\uparrow 6.34\%}$ | $0.376_{\uparrow 35.41\%}$ |
| PDF | $0.408_{\uparrow 2.00\%}$ | $0.345_{\uparrow 2.07\%}$ | $0.898_{\uparrow 18.46\%}$ | $0.268_{\uparrow 9.38\%}$ | $0.717_{\uparrow 10.99\%}$ | $0.291_{\uparrow 16.39\%}$ |
| CycleNet | $0.412_{\uparrow 3.00\%}$ | $0.344_{\uparrow 1.77\%}$ | $0.971_{\uparrow 28.10\%}$ | $0.269_{\uparrow 9.79\%}$ | $0.747_{\uparrow 15.63\%}$ | $0.310_{\uparrow 21.67\%}$ |
| SpareTSF | $0.400_{\uparrow 0.00\%}$ | $0.350_{\uparrow 3.55\%}$ | $1.556_{\uparrow 105.27\%}$ | $0.303_{\uparrow 23.67\%}$ | $0.858_{\uparrow 32.81\%}$ | $0.339_{\uparrow 28.31\%}$ |
| **PHAT (Ours)** | **0.400** | **0.338** | **0.758** | **0.245** | **0.646** | **0.243** |

Table 9: Comparison with various self-attention mechanisms The results are average MAE among all horizon.

| Datasets | ETTh | ETTm | ILI |
|---|---|---|---|
| + CrossFormer | 0.441 | 0.380 | 0.742 |
| + Vanilla Transformer | 0.424 | 0.366 | 0.768 |
| + PatchFormer | 0.435 | 0.373 | 0.825 |
| **PHAT (Ours)** | **0.400** | **0.338** | **0.758** |

aggregating multiple time steps into a single token. While this approach improves computational efficiency, it comes at the cost of mixing distinct temporal roles, leading to the loss of fine-grained alignment among the original time steps.

In contrast, our period folding approach preserves the full temporal resolution by avoiding any form of down-sampling. Instead, the sequence is reorganized into two orthogonal axes—period-offset and period-aligned—based on the detected periodic structure. Attention is applied along both

Table 10: Details of combination Ablation Study of the period-offset attention in PNA. Pos (Neg): Positive (Negative) components. Mod: Modulation.

| Variants | | ETTh1 | | ETTh2 | | ETTm1 | | ETTm2 | | NN5 | | FRED-MD | | AQShunyi | | ILI | | CzeLan | | NASDAQ | |
|---|---|---|---|---|---|---|---|---|---|---|---|---|---|---|---|---|---|---|---|---|---|
| Pos. | Neg. | MSE | MAE | MSE | MAE | MSE | MAE | MSE | MAE | MSE | MAE | MSE | MAE | MSE | MAE | MSE | MAE | MSE | MAE | MSE | MAE |
| - | - | 0.369 | 0.390 | 0.275 | 0.336 | 0.305 | 0.340 | 0.169 | 0.253 | 0.688 | 0.552 | 25.698 | 0.980 | 0.666 | 0.468 | 1.551 | 0.775 | 0.185 | 0.222 | 0.440 | 0.490 |
| - | ✓ | 0.368 | 0.386 | 0.273 | 0.331 | 0.302 | 0.339 | 0.168 | 0.254 | 0.688 | 0.551 | 25.305 | 0.954 | 0.665 | 0.468 | 1.531 | 0.765 | 0.176 | 0.221 | 0.433 | 0.483 |
| ✓ | - | 0.363 | 0.384 | 0.272 | 0.330 | 0.297 | 0.332 | 0.167 | 0.253 | 0.684 | 0.553 | 25.520 | 0.967 | 0.665 | 0.468 | 1.485 | 0.759 | 0.175 | 0.215 | 0.427 | 0.476 |
| ✓ | ✓ | 0.361 | 0.383 | 0.272 | 0.330 | 0.287 | 0.330 | 0.162 | 0.246 | 0.681 | 0.551 | 21.495 | 0.903 | 0.665 | 0.468 | 1.318 | 0.705 | 0.170 | 0.204 | 0.416 | 0.473 |
| PMod. | NMod. | MSE | MAE | MSE | MAE | MSE | MAE | MSE | MAE | MSE | MAE | MSE | MAE | MSE | MAE | MSE | MAE | MSE | MAE | MSE | MAE |
| - | - | 0.368 | 0.390 | 0.275 | 0.334 | 0.309 | 0.345 | 0.166 | 0.243 | 0.688 | 0.554 | 23.869 | 0.942 | 0.669 | 0.468 | 1.560 | 0.774 | 0.177 | 0.284 | 0.420 | 0.479 |
| - | ✓ | 0.365 | 0.386 | 0.272 | 0.330 | 0.305 | 0.339 | 0.167 | 0.242 | 0.685 | 0.553 | 24.740 | 0.943 | 0.666 | 0.468 | 1.535 | 0.764 | 0.174 | 0.252 | 0.419 | 0.478 |
| ✓ | - | 0.362 | 0.385 | 0.272 | 0.330 | 0.295 | 0.334 | 0.166 | 0.243 | 0.685 | 0.554 | 23.876 | 0.907 | 0.665 | 0.468 | 1.446 | 0.750 | 0.177 | 0.234 | 0.416 | 0.479 |
| ✓ | ✓ | 0.361 | 0.383 | 0.272 | 0.330 | 0.287 | 0.330 | 0.162 | 0.246 | 0.681 | 0.551 | 21.495 | 0.903 | 0.665 | 0.468 | 1.318 | 0.705 | 0.170 | 0.204 | 0.416 | 0.473 |

axes, enabling the model to explicitly capture rich intra-period and inter-period dependencies while maintaining the integrity of the original temporal sequence.

A related idea can be observed in TimesNet (Wu et al., 2022a), which also employs a folding operation. However, TimesNet extracts features from the folded representation using convolutional kernels. These kernels aggregate information across multiple positions, which can blur inherent periodic boundaries and disrupt latent cyclic correlations. This misalignment between convolutional receptive fields and true periodic structures limits TimesNet's ability to fully capture cycle-aware dependencies. In contrast, our model applies X-shaped attention directly to the folded representation, preserving periodic fidelity and ensuring that the periodic structure is consistently leveraged throughout the learning process.

In summary, our fold-and-attend approach stands apart by preserving the full temporal sequence and explicitly structuring it along periodic axes. Unlike patch-based Transformers, which sacrifice temporal resolution for efficiency, or TimesNet, which disrupts periodic integrity with convolutional receptive fields, our design respects periodicity at its core. This allows our model to generalize effectively across diverse temporal patterns while offering enhanced interpretability in capturing heterogeneous periodic dependencies.

# F   USE OF LLM

We only use LLMs as a language optimization tool to polish sentences, improving their readability and fluency.

