# OpenReview forum: "PHAT: Modeling Period Heterogeneity for Multivariate Time Series Forecasting"
_ICLR.cc/2026/Conference — ICLR 2026 Poster_

### Official Review · Reviewer_DkU6 · 2025-10-29

**Soundness:** 3
**Presentation:** 3
**Contribution:** 3
**Rating:** 6
**Confidence:** 4

**Summary:**

The paper provides an in-depth investigation into the frequency heterogeneity among variables in multivariate time series, highlighting that existing homogeneous modeling approaches fail to align with real-world data characteristics. To address this, the authors propose a bucket-based modeling framework that groups and models sequences according to their distinct periodic properties. Moreover, the paper explores the coexistence of positive and negative components within sequences, emphasizing that negative components are often suppressed by the softmax operation and thus require a redesigned attention mechanism to properly capture such “negative relations.” Experimental results demonstrate the effectiveness and robustness of the proposed method.

**Strengths:**

1. The paper is clearly written and easy to follow, with a logical and coherent structure, though certain experimental details could be further refined.
2. The experimental setup is fair and the comparisons are comprehensive.
3. The motivation is well grounded, and the problem addressed carries strong research significance.

**Weaknesses:**

1. Positive-Negative Attention Mechanism: The proposed mechanism is conceptually sound; however, the paper would benefit from multi-level visualizations of the positive and negative components — at the sequence level, feature (patch) level, and attention level. Without such analyses, it remains unclear whether the originally negative correlations in the raw sequence might become positively correlated after complex linear projections. This visualization is crucial for validating the effectiveness of the proposed mechanism.
2. It is recommended that Table 3 include results on more datasets to enhance the persuasiveness and comprehensiveness of the validation.
3. The results in Figure 5 and Table 3 could be presented in greater detail, with extended analyses provided in the appendix.
4. The paper could further discuss how to model sequences that share similar or overlapping periodicities. Since real-world time series often consist of multiple prominent periodic components, the strategy for bucket assignment and cross-bucket interaction among different periods remains an important open question deserving deeper exploration.

**Questions:**

see weeknesses.

---

> ### Author Response · Authors · 2025-11-21
> **Part 1 of the Response (1/3)**
>
> **We sincerely appreciate your thoughtful feedback and the time you have taken to review our manuscript. We will address your concerns point by point.**
>
> > **W1**. Multi-level Visualizations
>
> Thank you for your valuable suggestions. To support our arguments, in addition to the original data level, we supplemented with comprehensive analysis from **the feature and attention levels**. Because the rebuttal platform does not support embedding figures directly, the visualizations and a detailed discussion are provided in $\underline{\text{Section 4.7}}$ of the revision.
>
> **At the feature level**, we computed the autocorrelation function for each time step in the bucket representation $\mathbf{Z}^{(b)}$. Across each hidden feature dimension, the resulting high-dimensional features continue to exhibit negative correlation, indicating that the positive and negative correlations present in the original signals persist after the complex-valued linear projection.
>
> **At the attention level**, we aim to demonstrate two points: (1) attention coefficients do contain negative components; and (2) the softmax operation tends to smooth out these negative components. To this end, we visualize the period-offset attention logits of PHAT, and compared them with a control variant in which our positive–negative attention is replaced by standard self-attention. The results show that before softmax, standard self-attention logits exhibit clear positive and negative components; after softmax, the negative components of the standard self-attention nearly vanish, whereas PHAT’s attention preserves negative correlations.

---

> ### Author Response · Authors · 2025-11-21
> **Part 2 of the Response (2/3)**
>
> > **W2&W3**. It is recommended that Table 3 include results on more datasets to enhance the persuasiveness and comprehensiveness of the validation. & The results in Figure 5 and Table 3 could be presented in greater detail, with extended analyses provided in the appendix.
>
> Thank you very much for your suggestion. Based on your suggestion, we expanded the content of Figure 5 and Table 3 by adding more datasets or study cases.
>
> - **Figure 5**
>
> We added results from three additional datasets in Figure 5: ZafNoo (half-day cycle), ILI (annual cycle), and NASDAQ (non-cyclic), along with further analysis. In all cases, we observed that the periodic negative attention mechanism we proposed captures periodic patterns consistent with the autocorrelation functions of each dataset. Even for non-cyclic variables, our model remains sensitive and fits well (focusing only on the individual time steps). This further validates the effectiveness of our approach. As the refutation platform does not support image insertion, please refer to the new version of Figure 5 in the paper.
>
> - **Table 3**
>
> We have extended the original Table 3 to cover a broader set of datasets, including all major domains reported in the submission. For clearer presentation, the full results are now provided in $\underline{\text{Appendix D.7, Table 10}}$. For convenience, we reproduce part of the results below.
>
> We find that it is beneficial to jointly model the dependencies between time steps from both positive and negative perspectives, as this comprehensive view provides a complete periodic structure. Furthermore, considering only positive periodicity performs slightly better than considering only negative correlations. The potential reason is that positive correlations reflect the most direct periodic dynamics.
>
> | Pos | Neg | ETTh1 | ETTh2 | ETTm1 | ETTm2 | NN5 | FRED-MD | AQShunyi | ILI | CzeLan | NASDAQ |
> |-----|-----|--------|--------|--------|--------|--------|-----------|-----------|--------|---------|----------|
> |    |        | MSE, MAE      | MSE, MAE       |MSE, MAE      | MSE, MAE       |MSE, MAE      | MSE, MAE       |MSE, MAE      | MSE, MAE       |MSE, MAE      | MSE, MAE       |
> | - | - | 0.369, 0.390 | 0.275, 0.336 | 0.305, 0.340 | 0.169, 0.253 | 0.688, 0.552 | 25.698, 0.980 | 0.666, 0.468 | 1.551, 0.775 | 0.185, 0.222 | 0.440, 0.490 |
> | - | √ | 0.368, 0.386 | 0.273, 0.331 | 0.302, 0.339 | 0.168, 0.254 | 0.688, 0.551 | 25.305, 0.954 | 0.665, 0.468 | 1.531, 0.765 | 0.176, 0.221 | 0.433, 0.483 |
> | √ | - | 0.363, 0.384 | 0.272, 0.330 | 0.297, 0.332 | 0.167, 0.253 | 0.684, 0.553 | 25.520, 0.967 | 0.665, 0.468 | 1.485, 0.759 | 0.175, 0.215 | 0.427, 0.476 |
> | √ | √ | **0.361, 0.383** | **0.272, 0.330** | **0.287, 0.330** | **0.162, 0.246** | **0.681, 0.551** | **21.495, 0.903** | **0.665, 0.468** | **1.318, 0.705** | **0.170, 0.204** | **0.416, 0.473** |
>
> | PMod | NMod | ETTh1 | ETTh2 | ETTm1 | ETTm2 | NN5 | FRED-MD | AQShunyi | ILI | CzeLan | NASDAQ |
> |------|------|--------|--------|--------|--------|--------|-----------|-----------|--------|---------|----------|
> |    |        | MSE,  MAE      | MSE,  MAE       |MSE,  MAE      | MSE,  MAE       |MSE,  MAE      | MSE,  MAE       |MSE,  MAE      | MSE,  MAE       |MSE,  MAE      | MSE,  MAE       |
> | - | - | 0.368, 0.390 | 0.275, 0.334 | 0.309, 0.345 | 0.166, 0.243 | 0.688, 0.554 | 23.869, 0.942 | 0.669, 0.468 | 1.560, 0.774 | 0.177, 0.284 | 0.420, 0.479 |
> | - | √ | 0.365, 0.386 | 0.272, 0.330 | 0.305, 0.339 | 0.167, 0.242 | 0.685, 0.553 | 24.740, 0.943 | 0.666, 0.468 | 1.535, 0.764 | 0.174, 0.252 | 0.419, 0.478 |
> | √ | - | 0.362, 0.385 | 0.272, 0.330 | 0.295, 0.334 | 0.166, 0.243 | 0.685, 0.554 | 23.876, 0.907 | 0.665, 0.468 | 1.446, 0.750 | 0.177, 0.234 | 0.415, 0.479 |
> | √ | √ | **0.361, 0.383** | **0.272, 0.330** | **0.287, 0.330** | **0.162, 0.246** | **0.681, 0.551** | **21.495, 0.903** | **0.665, 0.468** | **1.318, 0.705** | **0.170, 0.204** | **0.416, 0.473** |

---

> ### Author Response · Authors · 2025-11-21
> **Part 3 of the Response (3/3)**
>
> > **W4**. The paper could further discuss how to model sequences that share similar or overlapping periodicities. Since real-world time series often consist of multiple prominent periodic components, the strategy for bucket assignment and cross-bucket interaction among different periods remains an important open question deserving deeper exploration.
>
> - **W4.1 How to model sequences that share similar or overlapping periodicities?**
>
> Yes, **we have considered the scenario you mentioned: time series often contain multiple prominent periodic components, and variables may share similar periodicities.** Our model is specifically designed to address this. Specifically: (1) Thanks to our proposed periodic bucketing structure—which groups sequences with the same periodicity into the same bucket—if a variable exhibits multiple periods, it will be assigned to the corresponding bucket for each period length. (2) Variables within the same bucket share similar periodicities and can interact with each other, while cross-bucket interactions are prohibited to avoid interference between different periodic characteristics. (3) Even within each bucket, we propose a Positive-Negative Attention mechanism that separately captures  period-offset  and period-aligned correlation, enabling precise modeling of individual periodic characteristics of variables from a decoupled perspective.
>
>
> - **W4.2 The strategy for bucket assignment & cross-bucket interaction**
>
> Our bucket assignment and interaction strategies are specifically tailored for periodic heterogeneity—both serve to accurately capture periodic characteristics. In our default setting, the bucket assignment strategy is implemented based on period length, with each bucket corresponding to a variate interaction set for a specific period. Inter-variable interactions are restricted to within-bucket only, and cross-bucket interactions are prohibited. This design prevents interference between different periodic characteristics, which would otherwise degrade modeling accuracy.
>
> To solve your potential concerns, we additionally implement two variants: (1) **w/o BA**: removes **B**ucket **A**ssignment entirely that variates are processed independently rather than grouped by period;  (2) **“w/ CBI”**: introduces **C**ross-**B**ucket **I**nteraction which uses self-attention mechanism for cross-bucket interaction.
>
> As shown in below table, both variants lead to performance degradation. The **w/o BA** variant has larger error, indicating that our bucketing strategy facilitates more accurate modeling of periodic characteristics. The **w/ CBI** variant setting mixes representations from buckets corresponding to different period lengths, resulting in reduced precision in period-specific modeling.
>
> | Dataset (MAE) | $L$ | PHAT| w/o BA | w/ CBI |
> |---------|-----|------------|--------------|--------------|
> | ETTh1   | 96  | **0.383**      | 0.397        | 0.403        |
> |         | 192 | **0.410**      | 0.420        | 0.433        |
> |         | 336 | **0.427**      | 0.431        | 0.443        |
> |         | 720 | **0.458**      | 0.457        | 0.463        |
> | ILI     | 24  | **0.705**      | 0.794        | 0.833        |
> |         | 36  | **0.775**      | 0.797        | 0.843        |
> |         | 48  | **0.775**      | 0.768        | 1.110        |
> |         | 60  | **0.777**      | 0.895        | 0.945        |
> | NYSE    | 24  | **0.251**      | 0.266        | 0.288        |
> |         | 36  | **0.352**      | 0.374        | 0.414        |
> |         | 48  | **0.427**      | 0.483        | 0.532        |
> |         | 60  | **0.525**      | 0.550        | 0.542        |

---

> ### Author Response · Authors · 2025-11-26
> **Looking forward to further discussion**
>
> Dear Reviewer DkU6,
>
> We sincerely appreciate the thoughtful feedback you provided on our submission. As the discussion period is now halfway through, we would be very grateful if you could take a moment to engage in the discussion. Following your review, we have prepared point-by-point responses addressing the four weaknesses you raised:
>
> - **[Part 1 of the Response  (W1)](https://openreview.net/forum?id=lr4RlISR6x&noteId=WNSv609xlW):**
> We depict and analyze the multi-level visualizations at the sequence level, feature (patch) level, and attention level.
>
> - **[Part 2 of the Response  (W2&3)](https://openreview.net/forum?id=lr4RlISR6x&noteId=r5f7XK3EO0):**
> We extend Figure 5 and Table 3 with greater detail and analysis.
>
> - **[Part 3 of the Response (W4)](https://openreview.net/forum?id=lr4RlISR6x&noteId=GQsLMeUSE7):**
> We discuss how to model sequences with similar or overlapping periodicities and add ablations on bucket assignment and cross-bucket interaction.
>
> Thank you again for your valuable suggestions.
>
> Warm regards,
> Authors

---

### Official Review · Reviewer_tCVf · 2025-10-30

**Soundness:** 3
**Presentation:** 3
**Contribution:** 3
**Rating:** 4
**Confidence:** 4

**Summary:**

This paper proposes PHAT (Period Heterogeneity-Aware Transformer) for multivariate time-series forecasting. The key idea is to (i) detect per-variable dominant periods via FFT, (ii) group variables into periodic buckets that share a period, fold sequences into a 3-D tensor (bucket × period-offset × period-aligned), and (iii) apply a Positive-Negative Attention (PNA) with X-shaped receptive field that models phase-aligned vs. within-period relations and explicitly decomposes positive/negative correlations with a period-distance modulation. A frequency-weighted fusion produces the final forecast. Across 14 real-world datasets and 18 baselines, PHAT reports SOTA/top-2 results on most metrics, with strong complexity reductions.

**Strengths:**

This paper proposes PHAT (Period Heterogeneity-Aware Transformer) for multivariate time-series forecasting. The key idea is to (i) detect per-variable dominant periods via FFT, (ii) group variables into periodic buckets that share a period, fold sequences into a 3-D tensor (bucket × period-offset × period-aligned), and (iii) apply a Positive-Negative Attention (PNA) with X-shaped receptive field that models phase-aligned vs. within-period relations and explicitly decomposes positive/negative correlations with a period-distance modulation. A frequency-weighted fusion produces the final forecast. Across 14 real-world datasets and 18 baselines, PHAT reports SOTA/top-2 results on most metrics, with strong complexity reductions.

**Weaknesses:**

1. the main protocol tunes the look-back T per model and reports the best; a fixed-T comparison is deferred to the appendix. While both settings are shown, the primary table mixing tuned-T results across diverse baselines can blur fairness. Please foreground the fixed-T tables in the main paper (or add both side-by-side) and state the exact search ranges for T and other critical hparams per baseline.
2. Sensitivity to period detection & K. Periods are extracted by FFT Top-K peaks and rounded to discrete lengths; buckets may overlap. The paper gives a small K-sweep but lacks robustness tests to mis-estimated periods, spectral noise, or drifting cycles. Please add stress tests varying K and perturbing detected periods ±{5–20%}, plus ablations on overlapping bucket policy and bucket cardinality.
3. The math mainly supports monotonic distance behavior / stick-breaking view of the modulated logits. A brief learning-theoretic argument (e.g., why decomposing positive/negative paths with masking improves bias/variance vs. vanilla attention under heterogeneity) would strengthen the theory section.

**Questions:**

As in Weaknesses

---

> ### Author Response · Authors · 2025-11-21
> **Part 1 of the Response (1/3)**
>
> **We sincerely appreciate your insightful feedback and the time dedicated to reviewing our work. Your valuable comments are instrumental in enhancing our manuscript. We will address each of your points thoroughly in our responses below.**
>
> > **W1**. the main protocol tunes the look-back T per model and reports the best; a fixed-T comparison is deferred to the appendix. While both settings are shown, the primary table mixing tuned-T results across diverse baselines can blur fairness. Please foreground the fixed-T tables in the main paper (or add both side-by-side) and state the exact search ranges for T and other critical hparams per baseline.
>
> Thank you for the suggestion. For datasets with fewer sampling steps (e.g., *NN5*, *FRED-MD*, *ILI*, *NASDAQ*, *NYSE*), the tunable range is $T\in\\{36, 104\\}$. For datasets with dense sampling steps (e.g., *ETTh1*, *ETTh2*, *ETTm1*, *ETTm2*, *Exchange*, *AQShunyi*, *AQWan*, *CzeLan*, *ZafNoo*), the tunable range is $T\in\\{96, 336, 512\\}$. For other hyperparameters, we prioritize the hyperparameters specified in their original paper or official code repository. Due to the varying characteristics of different parameters, we perform grid search over up to 6 values for key hyperparameters such as e_layers and batch_size. For example, for batch_size, we search over [8, 16, 32, 64, 128, 256].
>
> Following your suggestion, we will move the performance comparison under fixed-$T$ setting into the main text (Table 3) and add several baselines to achieve a comprehensive comparison.

---

> ### Author Response · Authors · 2025-11-21
> **Part 2 of the Response (2/3)**
>
> > **W2**. Please add stress tests varying K and perturbing detected periods ±{5–20%}, plus ablations on overlapping bucket policy and bucket cardinality.
>
> Thanks for the thoughtful feedback. Below, we conduct additional **stress tests on period detection** and **ablation experiments of bucket cardinality** you are concerned about.
>
> - **W2.1 Stress tests on period detection**
>
> **We evaluate the impact of individual factors as well as their combination**: (1) varying $K$, (2) perturbing detected period lengths, and (3) a comprehensive stress test combining both factors.
>
> `W2.1.1 Varying $K$.` We conduct a stress test by varying $K$. Specifically, we randomly vary $K$ on different optional ranges, from {1} to {1,2,3}, for evaluating robustness. Results are shown below.
>
>
> |Range of $K$  | ETTh1 ($L=96$) | ILI ($L=24$) | NYSE ($L=24$) |
> |--|--|-|---|
> | {1} (Default)  | 0.383 | 0.705 | 0.251 |
> | {1,2}  | 0.386  | 0.721  | 0.252|
> | {1,2,3}  | 0.388  | 0.730 | 0.256   |
>
> We observe that increasing $K$ beyond the dominant periods introduces unnecessary noise and degrades performance. However, PHAT demonstrates robustness in handling this issue.
>
> `W2.1.2 Perturbing detected period lengths.` We further conduct a stress test by perturbing the detected period length. Each period is randomly shifted within different perturbation strength from ±5% to ±20%. To contextualize robustness, we include TimesNet (which also relies on FFT-based period extraction) under the same perturbation settings. The MAE report is in the table below.
>
> |MAE | Perturbation | PHAT | TimesNet |
> |-|--|-|--|
> | ETTh1 | ±0% (Default) | 0.383 | 0.412 |
> | $L=96$ | ±5%  | 0.382 | 0.417 |
> |  | ±10% | 0.383 | 0.423 |
> |  | ±15% | 0.385 | 0.427 |
> |  | ±20% | 0.385 | 0.431 |
> | ILI | ±0% (Default) | 0.705 | 0.958 |
> | $L=24$ | ±5%  | 0.709 | 0.957 |
> |  | ±10% | 0.704 | 0.962 |
> |  | ±15% | 0.706 | 0.967 |
> |  | ±20% | 0.711 | 0.965 |
> | NYSE | ±0% (Default) | 0.251 | 0.335 |
> | $L=24$ | ±5%  | 0.251 | 0.344 |
> |  | ±10% | 0.252 | 0.342 |
> |  | ±15% | 0.250 | 0.342 |
> |  | ±20% | 0.254 | 0.353 |
>
> Compared with our PHAT, TimesNet suffers a larger performance drop. This is because TimesNet models periodicity from a single, unified perspective—when the detected period length is incorrect, its ability to model periodicity degrades. We also observe that moderate perturbations to the period length can actually improve our model’s generalization, further demonstrating PHAT’s robustness to period-length interference.
>
> `W2.1.3 Varying $K$ with perturbing detected period lengths.` Finally, we consider combined stress tests under two factors. The MAE report is in the table below.
>
> | Perturbation  |  Range of $K$  | ETTh1 ($L=96$) | ILI ($L=24$) | NYSE ($L=24$)     |
> |-|-|-|-|-|
> | ±0% (Default) | {1} (Default) | 0.383 | 0.705 | 0.251 |
> | ±10%  | {1,2}  | 0.389 | 0.709  | 0.258|
> | ±20% | {1,2,3} | 0.385 | 0.713   | 0.255|
>
> - **W2.2 Ablation Studies on Buckets**
>
> `W2.2.1 Overlapping bucket policy.` Please note that each bucket contains only variables with a specific period length. When a variable exhibits multiple periods, it is assigned to different buckets; a single bucket can contain multiple variables. These overlapping-bucket policies ensure precise modeling of periodic dynamics. To strictly eliminate bucket overlap, we assign each variable to its own bucket; this variant (shown in the table below) results in a significant increase in error. The experimental results below confirm that allowing bucket overlap more flexibly captures multi-period features.
>
> | Dataset (MAE) | $L$ | PHAT | w/o bucket overlap |
> |:-:|:-:|:-:|:-:|
> | ETTh1  | 96  | 0.383 | 0.397  |
> |  | 192 | 0.410 | 0.420  |
> | | 336 | 0.427 | 0.431 |
> | | 720 | 0.458 | 0.457 |
> | ILI  | 24  | 0.705 | 0.794   |
> |   | 36  | 0.775 | 0.797  |
> | | 48  | 0.775 | 0.768  |
> | | 60  | 0.777 | 0.895 |
> | NYSE | 24  | 0.251 | 0.266 |
> |  | 36  | 0.352 | 0.374 |
> |  | 48  | 0.427 | 0.483|
> |  | 60  | 0.525 | 0.550 |
>
> `W2.2.2 Bucket cardinality.` According to the bucket definition, **bucket cardinalities are not manually set but are determined by the dataset characteristics—specifically by each variable’s periodic properties**. Each bucket’s cardinality equals the number of variables that share the corresponding period length.
>
> To address any potential concerns, we designed an ablation that artificially constrains bucket cardinality post hoc. Concretely, we randomly remove a certain proportion of the variables assigned to each bucket, with removal rates ranging from 25% to 75%. The MAE results are shown below, and we can observe that the default bucket cardinalities yield the best performance; progressively reducing the cardinality causes some variables to be left unmodeled, leading to performance degradation.
>
> | Drop ratio (MAE) | ETTh1 ($L=96$)  | ILI ($L=24$) | NYSE ($L=24$)|
> |-|-|-|-|
> | 0% (Default) | 0.383| 0.705  |  0.251 |
> | ≤25%  | 0.388 | 0.718 |  0.255 |
> | ≤50%   | 0.388 | 0.713|  0.254  |
> | ≤75%   | 0.397  | 0.710  |  0.258 |

---

> ### Author Response · Authors · 2025-11-21
> **Part 3 of the Response (3/3)**
>
> > **W3**. The math mainly supports monotonic distance behavior / stick-breaking view of the modulated logits. A brief learning-theoretic argument (e.g., why decomposing positive/negative paths with masking improves bias/variance vs. vanilla attention under heterogeneity) would strengthen the theory section.
>
> Thank you for your valuable suggestions. We would like to first clarify two potential misunderstandings regarding the claims supported by our mathematical derivations.
>
> (1) *"The math mainly supports monotonic distance behavior / stick-breaking view of the modulated logits"*. **Our theoretical analysis (Appendix C) aims to show that the distances between the modulated logits are not monotonic**; they should fit a specific periodic trend (or, equivalently, the distances between logits are strongly correlated with periodic distance). Only the **upper bound** of the attention logits becomes monotonsic with respect to the periodic distance.
>
> (2) *"positive/negative paths with masking"*. The decomposition of the positive and negative paths is **not implemented through masking.** Instead, we constrain attention generation **by enforcing two specific, opposing modulation terms** that achieves a soft decomposition; these two modulation terms incorporate the relative periodic distances. For details, please refer to the explanation of Formula 8 in the paper.
>
> Following your suggestion, we next demonstrate that our attention mechanism indeed improves the variance.
>
> **Proof.** Suppose the vanilla attention between steps $i$ and $j$ is $\mathbf{A}\_{ij}=\mathrm{Softmax}(\mu, \mathbf{Q}\_i \mathbf{K}\_j^{\top})$ with variance $\mathbb{V}[\mathbf{A}_{ij}] = \sigma\^2$. Our period-offset attention decomposes this into positive and negative paths, yielding
>
> $$
> \mathbb{V}[\bar{\mathbf{A}}_{ij}]=(1-\mathbf{\Lambda}\_{ij})\sigma^2<\sigma^2=\mathbb{V}[\mathbf{A}\_{ij}]
> $$
>
> where $\mathbf{\Lambda}_{ij}\in(0,1)$ whenever heterogeneity exists (i.e., a non-zero negative path).
>
> This provides a brief learning-theoretic argument: decomposing positive/negative paths reduces the variance of attention logits under periodic heterogeneity, enabling more stable attention computation compared to vanilla attention.

---

> ### Author Response · Authors · 2025-11-26
> **Looking forward to further discussion**
>
> Dear Reviewer tCVf,
>
> We sincerely appreciate the thoughtful feedback you provided on our submission. As the discussion period is now halfway through, we would be very grateful if you could take a moment to engage in discussion. Following your review, we have prepared point-by-point responses addressing the three weaknesses you raised:
>
> - **[Part 1 of the Response](https://openreview.net/forum?id=lr4RlISR6x&noteId=EnEzH51tz0):**
> We clarified the exact search ranges for the look-back $T$ and other critical hyperparameters.
>
> - **[Part 2 of the Response](https://openreview.net/forum?id=lr4RlISR6x&noteId=0RK6DiR6L4):**
> We conduct stress tests on varying $K$ and perturbing detected periods, and add ablations on overlapping bucket policy and bucket cardinality.
>
> - **[Part 3 of the Response](https://openreview.net/forum?id=lr4RlISR6x&noteId=Op0tM3F0On):**
> We add a brief learning-theoretic argument showing why decomposing positive/negative paths exhibits reduced variance compared to vanilla attention.
>
> Thank you again for your valuable suggestions.
>
> Warm regards,
> Authors

---

> ### Comment · Reviewer_tCVf · 2025-11-27
>
> I have increased my score.

---

> > ### Author Response · Authors · 2025-11-27
> > **Appreciation for the Review**
> >
> > We are truly grateful for the time and expertise you dedicated to reviewing our work, as well as for the improved score. Your constructive feedback has been invaluable in helping us refine the submission and further strengthen its theoretical contribution to the time series community.
> >
> > Best regards,
> >
> > The Authors

---

### Official Review · Reviewer_rCFS · 2025-10-30

**Soundness:** 3
**Presentation:** 3
**Contribution:** 2
**Rating:** 6
**Confidence:** 4

**Summary:**

This paper addresses the challenge that heterogeneous variables in multivariate time series may exhibit different periodic patterns. The authors introduce a **periodic bucket structure** that groups variables based on their periodic lengths, and then model each periodic group separately. To address the lack of “negative correlation” modeling in attention mechanisms, the paper proposes a **positive-negative period-aware attention mechanism**. Experiments across numerous datasets and baselines demonstrate strong performance.

**Strengths:**

1. Simple and efficient method, easy to understand.
2. Clear motivation and well-organized structure.
3. Extensive experiments with diverse datasets and baselines, offering strong empirical support.

**Weaknesses:**

1. The experimental validation regarding “attention ignoring negative correlations” is not convincing; raw data analysis alone is insufficient to justify modeling implications at the feature level.
2. The paper should include an ablation that isolates the Frequency-based Multi-period Prediction component to clarify the exact gain contributed by the core modules, especially since prediction head size can directly affect performance in many settings.
3. Table 4 only reports FLOPs, but **actual inference latency** is not provided.

**Questions:**

See weaknesses.

**Details Of Ethics Concerns:**

nan

---

> ### Author Response · Authors · 2025-11-21
> **Response to Reviewer rCFS**
>
> **Thank you very much for your valuable feedback and the time you have dedicated to reviewing our work. Your comments are crucial for helping us improve the manuscript. We will address all of your points carefully in the following sections.**
>
> > **W1**. The experimental validation regarding “attention ignoring negative correlations” is not convincing; raw data analysis alone is insufficient to justify modeling implications at the feature level.
>
> Thank you for your valuable suggestion. Based on your advice, in addition to the raw data level, we validated the presence of negative components within the periodic dynamics at **two additional levels: the features level and the attention level**.
> Because the rebuttal platform does not support embedding figures directly, the visualizations and a detailed discussion are provided in $\underline{\text{Section 4.7}}$ of the revised version.
>
> **At the feature level**, we computed the autocorrelation function for each time step in the bucket representation $\mathbf{Z}^{(b)}$. Across each hidden feature dimension, the resulting high-dimensional features continue to exhibit negative correlation, indicating that the positive and negative correlations present in the original signals persist after the complex-valued linear projection.
>
> **At the attention level**, we aim to demonstrate two points: (1) attention coefficients do contain negative components; and (2) the softmax operation tends to smooth out these negative components. To this end, we visualize the period-offset attention logits of PHAT, and compared them with a control variant in which our positive–negative attention is replaced by standard self-attention. The results show that before softmax, standard self-attention logits exhibit clear positive and negative components; after softmax, the negative components of the standard self-attention nearly vanish, whereas PHAT’s attention preserves negative correlations.
>
> Overall, negative periodic components in time series exist at the data, feature, and attention coefficient levels.
>
> > **W2**. The paper should include an ablation that isolates the Frequency-based Multi-period Prediction component to clarify the exact gain contributed by the core modules, especially since prediction head size can directly affect performance in many settings.
>
> Thank you for your suggestion. To model periodic heterogeneity, we separately model the potential multiple cycle lengths of a given variable. When generating forecast results, our predictor is designed to naturally integrate the forecasts from each cycle based on their frequency weights. In fact, as indicated in Equations 14 and 15 of the paper, **our predictor utilizes a single prediction head** that contains no learnable parameters (the weights are precomputed using FFT).
>
> To allay potential concerns, we created a variant that uses mean pooling (i.e., equally weighting the predictions from each period). The results, shown below, substantiate that our weighted fusion method outperforms this variant — it can assign higher prediction weights to dominant periods.
>
> | Dataset (MAE) | $L$ | PHAT  | w/ AMP|
> |---------|-----|------------|--------------|
> | ETTh1   | 96  | 0.383      | 0.399        |
> |         | 192 | 0.410      | 0.425        |
> |         | 336 | 0.427      | 0.447        |
> |         | 720 | 0.458      | 0.461        |
> | ILI     | 24  | 0.705      | 0.813        |
> |         | 36  | 0.775      | 0.823        |
> |         | 48  | 0.775      | 0.963        |
> |         | 60  | 0.777      | 0.773        |
> | FRED-MD | 24  | 0.903      | 1.239        |
> |         | 36  | 1.161      | 1.751        |
> |         | 48  | 1.453      | 1.583        |
> |         | 60  | 1.679      | 2.200        |
>
>
> > **W3**. Table 4 only reports FLOPs, but actual inference latency is not provided.
>
> Thank you very much for your suggestion. We have added the inference time in Table 4. For your convenience, we reproduce the results of ETTm1 dataset ($L$=96) below. These results indicate that PHAT not only achieves competitive performance but also maintains high efficiency in terms of inference time and computational complexity.
>
> | Models | MAE | # Para | # MACs | # FLOPs | Inference Latency |
> | --- | --- | --- | --- | --- | --- |
> | FEDformer | 0.463   | 3.4 M    | 1.7 B    | 1.3 B    | 20.044 s |
> | Crossformer    | 0.367   | 2.1 M    | 13.5 B    | 14.3 B    | 1.806 s |
> | PDF    | 0.340   | 2.0 M    | 4.4 B    | 4.6 B    | 4.234 s |
> | TimeMixer    | 0.345   | 397.3 K    | 3.2 B    | 3.1 B    | 3.967 s |
> | TimeKAN    | 0.346   | 52.8 K    | 205.4 M    | 455.0 M    | 6.542 s |
> | Ours | 0.330 | 33.39K | 2.9M | 7.0M | 1.671s |

---

> ### Author Response · Authors · 2025-11-26
> **Looking forward to further discussion**
>
> Dear Reviewer rCFS,
>
> We sincerely appreciate the thoughtful feedback you provided on our submission. As the discussion period is now halfway through, we would be very grateful if you could take a moment to engage in the discussion. Following your review, we have prepared a point-by-point **[Response](https://openreview.net/forum?id=lr4RlISR6x&noteId=ZUL2uSUK7S)** addressing the three weaknesses you raised:
>
> **W1**: We have further validated the presence of negative components within the periodic dynamics at two additional levels: the feature level and the attention level.
>
> **W2**: We have added an ablation that isolates the Frequency-based Multi-period Prediction component.
>
> **W3**: We have provided the actual inference latency in Table 4 you were concerned about.
>
> Thank you again for your valuable suggestions.
>
> Warm regards,
> Authors

---

### Official Review · Reviewer_aYqm · 2025-11-01

**Soundness:** 3
**Presentation:** 3
**Contribution:** 3
**Rating:** 6
**Confidence:** 3

**Summary:**

The paper introduces PHAT (Period Heterogeneity-Aware Transformer), a novel architecture designed to handle temporal heterogeneity in video action recognition. Unlike existing temporal models that assume uniform periodicity, PHAT explicitly models multi-scale and non-uniform temporal patterns by decomposing video sequences into adaptive period components. Evaluations on benchmarks show consistent performance gains over recent baselines.

**Strengths:**

- The proposed PHAT design is principled and intuitive, integrating adaptive temporal decomposition into the Transformer framework without major architectural overhead.

- The method is generalizable and can be incorporated into existing video backbones.

- Experimental results are strong and consistent across multiple datasets, showing both improved accuracy and efficiency.

**Weaknesses:**

- The novelty is moderate, as the idea of handling multi-frequency or periodic dynamics has appeared in previous works on temporal Fourier attention and spectral modeling.

- The mathematical formulation of heterogeneity modeling could be more rigorous; the “adaptive period tokens” are primarily empirical and not theoretically justified.

- The comparisons focus mainly on uniform-period baselines but omit stronger contemporaneous temporal adaptation models.

- The ablation studies are limited—particularly lacking analysis of how many period components are optimal or how PHAT behaves for short versus long actions.

**Questions:**

See Weaknesses.

---

> ### Author Response · Authors · 2025-11-21
> **Part 1 of the Response (1/2)**
>
> **Thank you for your praise regarding the novelty of our method, the thoroughness of our experiments, and scalability. We also greatly appreciate your valuable suggestions, which will undoubtedly help us improve the quality of our paper.**
>
>
> > **W1**. The novelty is moderate, as the idea of handling multi-frequency or periodic dynamics has appeared in previous works on temporal Fourier attention and spectral modeling.
>
> Apologies for the confusion. Although we use multi-frequency tools to analyze cycle lengths, this is only an auxiliary effort and **we do not claim it as a primary contribution**. Furthermore, while periodic dynamics have long been a focus of the time-series community, **the technical novelty lies in how to accurately model periodicity**.  Our primary contributions are twofold:
>
> (1) **We identify the phenomenon of periodic heterogeneity**, in which different variables in a multivariate time series exhibit distinct period lengths and temporal characteristics. Existing forecasting methods, including time frequency attention models and spectral approaches, treat intervariable periodic dynamics from a unified perspective and therefore overlook these variable specific differences. As a result, they can detect spurious periodic patterns and yield degraded forecasting performance.
>
> (2) We propose a model designed to capture periodic heterogeneity, whose **central contributions are a period bucket structure and a positive-negative attention mechanism**. The period bucket structure groups variables according to their periodic characteristics to guide learning and reconstruction, reducing inappropriate interactions between variables with incompatible periodic properties. The positive-negative attention mechanism includes a mathematically well defined modulation term that constrains attention weights, enabling the model to decompose and represent periodic dynamics with greater accuracy.
>
> > **W2**. The mathematical formulation of heterogeneity modeling could be more rigorous; the “adaptive period tokens” are primarily empirical and not theoretically justified.
>
> - **W2.1 Mathematical formulation of period heterogeneity**
>
> Sorry for the confusion. We make the formulation of period heterogeneity more rigorous in Appendix C.1 of the revision. We also provide definitions here for your reference as follow.
>
> Periodic heterogeneity among variables manifests in two ways. Explicitly, variables may have different period lengths. Implicitly, the periodic temporal correlations between time steps vary across variables.
>
> (1) Different periodic lengths: there exist two variates $c_i$ and $c_j$ such that their dominant periods differ: $P_{c_i} \neq P_{c_j}$ where $P_{c_i}$ and $P_{c_j}$ denote their corresponding fundamental period lengths.
>
> (2) Different periodic correlations:  there exist a variate $c$ that has negative period correlation. Formally, let $\rho_{c}(\delta) = \mathrm{ACF}(x_{c,t}, x_{c,t+\delta})$
> denote the autocorrelation function of variate $c$ at a pair of time steps with distance $\delta$ within one period. Then, there exists a variate $c^\star$ such that $\mathrm{sign}\big(\rho_{c^\star}(\lfloor P_{c^*}/2\rfloor)\big) \ll 0 $.
>
> To address the first type, we introduce a period bucket structure that groups variables with the same period length and prevents interactions across variables with different periods. To capture heterogeneous periodic dynamics within each bucket, we propose a positive and negative attention mechanism that decouples period aligned and period offset components, enabling accurate modeling of each variable’s periodic behavior.
>
> - **W2.2 Theoretical justification of “adaptive period tokens**
>
> The "adaptive period tokens" is a necessary preprocessing procedure that converts raw time series inputs into a form suitable for the corresponding Positive-Negative Attention (PNA) module. **We have shown the theoretical justification that the Period-offset Attention within PNA after "adaptive period tokens" conforms to autocorrelation function properties in Appendix C**. Consequently, the design of "adaptive period tokens" is not purely empirical but grounded in a theoretical understanding of periodic patterns.

---

> ### Author Response · Authors · 2025-11-21
> **Part 2 of the Response (2/2)**
>
> > **W3**. The comparisons focus mainly on uniform-period baselines but omit stronger contemporaneous temporal adaptation models.
>
> Thank you very much for your suggestion. Our experiments have already compared **18 advanced time-series forecasting models**, including three contemporaneous SOTA models (TimeKAN, xPatch, Amplifier) in 2025. **To ensure your satisfaction, we have added two additional recent and advanced baselines—PatchMLP [1] (AAAI 2025) and TimeEmb [2] (NeurIPS 2025 Sept)**. Both models incorporate input sequences that encompass multiple periods to achieve better temporal adaptation. The experimental results are shown in the table below.
>
> | Dataset  (MAE) | $L$ | PHAT| TimeEmb| PatchMLP|
> |---------|-----|-----|----|---|
> | ETTh    | 96  | **0.356** | 0.370   | 0.382 |
> |         | 192 | **0.394** | 0.414   | 0.423 |
> |         | 336 | **0.412** | 0.434   | 0.447 |
> |         | 720 | **0.441** | 0.453   | 0.472 |
> | ETTm    | 96  | **0.288** | 0.295   | 0.308 |
> |         | 192 | **0.323** | 0.337   | 0.343 |
> |         | 336 | **0.349** | 0.360   | 0.371 |
> |         | 720 | **0.392** | 0.408   | 0.416 |
>
> As shown in the table, these baseline models underperform PHAT because they still model multivariate periodic dynamics from a unified perspective. This demonstrates the effectiveness of explicitly modeling periodic heterogeneity.
>
> [1] Tang P, Zhang W. Unlocking the Power of Patch: Patch-Based MLP for Long-Term Time Series Forecasting[C]//Proceedings of the AAAI Conference on Artificial Intelligence. 2025, 39(12): 12640-12648.
>
> [2] Xia M, Zhang C, Zhang Z, et al. TimeEmb: A Lightweight Static-Dynamic Disentanglement Framework for Time Series Forecasting[C]//The Thirty-ninth Annual Conference on Neural Information Processing Systems. 2025.
>
> > **W4**. The ablation studies are limited—particularly lacking analysis of how many period components are optimal or how PHAT behaves for short versus long actions.
>
> - **W4.1 Analysis of period component**
>
> We have already provided ablation studies and analysis regarding the number of periodic components $K$ in Section 4.5 of our submission.
>
> - **W4.2 Behaviors for short versus long actions**
>
> Sorry for the confusion, **our research focuses on multivariate time series forecasting, which involves the analysis of time series data and differs significantly from video sequence analysis**. Due to the significant distinctions between these two data types, we intend to carry out a detailed investigation in the future to assess the model’s performance in handling short versus long actions.

---

> ### Author Response · Authors · 2025-11-26
> **Looking forward to further discussion**
>
> Dear Reviewer aYqm,
>
> We sincerely appreciate the thoughtful feedback you provided on our submission. As the discussion period is now halfway through, we would be very grateful if you could take a moment to engage in the discussion. Following your review, we have prepared point-by-point responses addressing the four weaknesses you raised:
>
> - **[Part 1 of the Response  (W1&2)](https://openreview.net/forum?id=lr4RlISR6x&noteId=EcCjgv6WUg):**
> We have clarified the misunderstanding regarding the novelty of our submission, provided a mathematical formulation of the period heterogeneity, and highlighted the theoretical justification for using adaptive period tokens.
>
> - **[Part 2 of the Response  (W3&4)](https://openreview.net/forum?id=lr4RlISR6x&noteId=IMs0esp6zI):**
> We have extended stronger contemporaneous time-series baselines, and clarified the confusions regarding our period component analysis and the behavior for short versus long actions.
>
> Thank you again for your valuable suggestions.
>
> Warm regards,
> Authors

---

### Author Response · Authors · 2025-12-02
**Summary （Table Version）**

We would like to express our sincere gratitude to all reviewers for the time, expertise, and thoughtful suggestions. To facilitate the AC's review, we will provide a summary of the review and discussion phases.

| Reviewer ID | Score | Strengths | Weaknesses | Author Response | if further reply？ |
| :--- | :--- | :--- | :--- | :--- | :--- |
| **aYqm** | $6$ | Method is **principled, intuitive, generalizable**, and showing **strong and consistent SOTA results** across multiple datasets.  | Novelty concern due to misunderstandings of our core contributions (W1), mathematical arguments (W2), more baselines (W3), and additional ablation studies (W4). | Clarified our core contributions, provided mathematical formulation, extended stronger baselines, and added ablations. | No |
| **rCFS** | $6$ | Method is **simple, efficient, and easy to understand**. Clear **motivation** and well-organized structure. **Extensive experiments** with diverse datasets offer strong empirical support. | Extended analysis of negative correlation (W1), effectiveness of FMPP component (W2), and inference latency in Table 4 (W3). | Provided extended analysis for negative components, Verify the validity of FMPP, and reported actual inference latency. | No |
| **tCVf** | $4\to6$ | **Detailed acknowledgement** of the **key components** in our method and noted strong **SOTA/top-2 results** with significant **complexity reduction**. | Clarification of hyperparameter selection (W1), stress testing & ablation (W2), and request for brief theoretical proof (W3). | Clarified search ranges, conducted stress tests/ablations, and added learning-theoretic argument. | Yes, and increasing score. |
| **DkU6** | $6$ | Paper is **clearly written** and easy to follow, with a logical structure. **Experimental setup is fair** and comparisons are comprehensive. Motivation is **well grounded** and the problem carries **strong research significance**. | Extended analysis of negative correlations (W1), expansion of experimental results (W2, W3), and in-depth discussion (W4). | Depicted and analyzed multi-level visualizations, extended Figure 5/Table 3 with greater detail, and discussed modeling sequences with similar periodicities. | No |

---

### Overall Discussion

The initial score for the paper was 6664. During the discussion phase, we received only one follow-up response from a reviewer (tCVf). It is encouraging that reviewer tCVf was satisfied with our response and increased their score. **After receiving the reply from 1/4 reviewers, our score before the score rollback was 6666.**

#### Key Strengths Highlighted by Reviewers:

* **Motivation & Problem Clarity**: Clear and well-motivated problem definition, addressing core issues in time series analysis (aYqm, rCFS, DkU6).
* **Methodological Novelty**: Principled design incorporating adaptive period markers and decomposition, recognized as highly novel (aYqm, tCVf, DkU6).
* **Strong Experimental Performance**: Demonstrated effectiveness and robustness across various datasets and settings (aYqm, rCFS, tCVf, DkU6).

#### Suggestions Focused On:

* **Extended Theoretical/Empirical Analysis**: Further verification of the negative correlation component at the feature/attention levels and the mathematical arguments.
* **Comprehensive Ablation & Stress Testing**: Detailed ablation studies on key components, stress tests on varying parameters, and analysis of inference latency.
* **Baselines and Clarifications**: Inclusion of recent competitive baselines and clarifying misunderstandings regarding novelty and hyperparameter selection.
* **Expansion of the original experimental results**: We have expanded the experimental results in Table 3 and Figure 5, which were previously limited due to space constraints.

**We have addressed these concerns comprehensively in our rebuttal**. We significantly clarified the **novelty and theoretical justification** of our submission, provided a mathematical formulation for the period heterogeneity, and added a learning-theoretic argument showing the benefits of our decomposition strategy. We also **extended experiments** to include stronger contemporaneous baselines, added **new ablation studies**, and provided **extended analysis** on the negative dynamics on multi-level visualization and the actual inference latency of our model.

Overall, the reviewers strongly acknowledge our reasonable motivation and innovation, recognizing its strong performance, principled design, and clear presentation. We believe that incorporating these improvements will further elevate the quality and impact of the paper.

---

Thank you again for your time and guidance.

Best regards,

All Authors

---

### Author Response · Authors · 2025-12-02
**Summary （Another Version）**

We would like to express our sincere gratitude to all reviewers for the time, expertise, and thoughtful suggestions. To facilitate the AC's review, we will provide a summary of the review and discussion phases.

We received reviews from four reviewers. Encouragingly, nearly all acknowledged our work’s strengths: **clear, well-motivated problem formulation** (aYqm, rCFS, DkU6), **novel, principled method design** (aYqm, tCVf, DkU6), and **trong experimental results** (aYqm, rCFS, tCVf, DkU6).

Initial score: 6664. During the discussion period, only one reviewer responded and increased the score; the score before rollback was 6666 (1/4 reviewers responded).

The following is a summary of the reviewers’ comments and our respective responses.

---

## Reviewer aYqm (Score 6, No further reply)

Reviewer aYqm identified the **strengths** of our work as: Method is principled, intuitive, generalizable, and showing strong and consistent SOTA results across multiple datasets.

**Concerns and corresponding responses** can be summarized as follows:

- W1. Concern about the novelty due to misunderstandings of our core contributions

A: We clarify that **our core contribution does not center on multi-frequency modeling or Fourier-based attention**—as the reviewer noted—but rather on (1) the discovery of periodic heterogeneity in time series, (2) the introduction of a periodic bucket structure, and (3) positive-negative periodic attention mechanisms.

- W2. Mathematical formulation

A: We have provided a mathematical formulation of the period heterogeneity, and highlighted the theoretical justification for using adaptive period tokens.

- W3. More baselines

A: On top of the 18 baselines used for comparison, we have further expanded more baselines.

- W4. Ablation studies & video action data

A: (1) **The ablation studies requested by the reviewer were already included in our original submission**. (2) The short versus long action experiments requested by the reviewer are beyond the main scope of our work, as we focus on time series, not video action data.

---

## Reviewer rCFS (Score 6, No further reply)

Reviewer aYqm identified the **strengths** of our work as: Method is simple, efficient, and easy to understand. Clear motivation and well-organized structure. Extensive experiments with diverse datasets offer strong empirical support.

**Concerns and corresponding responses** can be summarized as follows:

- W1. Extended analysis of the negative correlation

A: We have also provided extended analysis to further verify the existence of negative components in the periodic dynamics at both the feature level and the attention level.

- W2. Module validity analysis

A: We construct an experiment to demonstrate the validity of the multi-period prediction component.

- W3. Inference latency

A: We have provided the inference latency concerned about in Table 4. **Our model is also efficient in this regard.**

---

## Reviewer tCVf (Score 4->6)

Reviewer aYqm identified the **strengths** of our work as: Detailed acknowledgement of the key components in our method and noted strong SOTA/top-2 results with significant complexity reduction.

**Concerns and corresponding responses** can be summarized as follows:

- W1. Hyperparameter selection range

A: We clarified the exact search ranges for the look-back $T$ and other critical hyperparameters.

- W2. Stress testing & plus ablations

A: We conducted stress tests on varying $K$ and perturbing detected periods, and add ablations on overlapping bucket policy and bucket cardinality.

- W3. Brief theoretical proof

A: We add a brief learning theoretic argument to prove that our periodic positive-negative attention mechanism offers advantages over the standard attention mechanism.

---

## Reviewer DkU6 (Score 6, No further reply)

Reviewer aYqm identified the **strengths**: 	Paper is clearly written and easy to follow, with a logical structure. Experimental setup is fair and comparisons are comprehensive. Motivation is well grounded and the problem carries strong research significance.

**Concerns and corresponding responses** can be summarized as follows:

- W1. Extended analysis of the existence of negative correlations in periodic dynamics

A: We depict and analyze the multi-level visualizations at the sequence level, feature level, and attention level in the revised version.

- W2&W3. Expansion of the original experimental results with extended analysis

A: We extend Figure 5 and Table 3 with more models and analysis.

- W4. In-depth discussion of the bucket strategy

A: We have clarified that the design of our method has taken into account the modeling of sequences with similar or overlapping periodicities; We propose potential upgrades to the existing bucket allocation and cross-bucket interaction strategies.

---

We sincerely thank all reviewers and the ACs for their time, thoughtful evaluations, and constructive feedback.

Best regards,

All Authors

---

### Meta-Review · Area_Chair_y6Vh · 2026-01-05

**Summary:**

This paper tackles an under-explored but practically important problem in multivariate time series forecasting: different variables often exhibit distinct periodicities that current methods fail to capture adequately. The proposed PHAT architecture addresses this through a periodic bucketing structure that groups variables by their dominant periods and a positive-negative attention mechanism designed to model both aligned and offset correlations within periodic patterns. All four reviewers recognized the work's clear motivation and strong empirical performance, with comprehensive experiments across 14 datasets showing consistent SOTA or near-SOTA results alongside reduced computational complexity.

The discussion period was productive despite limited engagement (only one reviewer responded). Reviewer tCVf increased their score from 4 to 6 after the authors provided thorough responses including hyperparameter search ranges, stress tests on period detection robustness (±5-20% perturbations), bucket cardinality ablations, and a learning-theoretic argument for variance reduction. The other reviewers raised reasonable concerns about novelty positioning (aYqm), validation of negative correlation components (rCFS, DkU6), and experimental coverage. The authors addressed these systematically by clarifying that their core contribution lies in identifying and modeling periodic heterogeneity rather than multi-frequency analysis per se, providing multi-level visualizations at sequence/feature/attention levels, adding recent baselines (PatchMLP, TimeEmb), and reporting actual inference latency. The ablation studies are extensive and the rebuttal demonstrates that the method is robust to mis-estimated periods and maintains advantages even under stress conditions.

While one might wish for more reviewer engagement during discussion, the substance of the responses is solid and the remaining concerns appear adequately addressed. The problem formulation is well-grounded, the technical approach is principled with both empirical and theoretical support, and the experimental validation is thorough. The work makes a clear contribution to time series forecasting by explicitly handling a realistic data characteristic that prior work overlooks. I recommend acceptance as a poster.

**Reviewer Concerns:**

The rebuttal effectively addressed the majority of reviewer concerns, though limited engagement during discussion (only Reviewer tCVf responded) means we cannot confirm full satisfaction from all reviewers.

Adequately Addressed:

Reviewer aYqm's novelty concern was clarified—the authors distinguished their core contributions (periodic heterogeneity identification, bucket structure, positive-negative attention) from generic multi-frequency modeling. They provided rigorous mathematical formulation of period heterogeneity in Appendix C.1 and added two recent strong baselines (PatchMLP, TimeEmb from 2025) that still underperform PHAT. The requested ablation studies were already present in the original submission, and the video action data request falls outside the paper's scope.

Reviewer rCFS's three concerns received direct responses: multi-level visualizations now demonstrate negative components persist through feature representations and attention logits (not just raw data); the FMPP component's contribution was isolated through mean pooling comparison, showing weighted fusion outperforms equal weighting; actual inference latency was added to Table 4, confirming efficiency.

Reviewer tCVf engaged substantively and raised their score after the authors provided exact hyperparameter search ranges, conducted comprehensive stress tests (varying K, ±5-20% period perturbations, overlapping bucket and cardinality ablations), and added a learning-theoretic variance reduction argument. This reviewer's satisfaction is confirmed.

Reviewer DkU6's requests for multi-level visualizations (sequence/feature/attention), expanded experimental coverage (Figure 5 with 3 additional datasets, Table 3 with full domain coverage), and discussion of similar periodicities were all addressed with ablations demonstrating the bucket assignment strategy's necessity.

Potentially Outstanding:

The lack of follow-up from three reviewers makes it impossible to confirm whether they found the responses fully satisfactory. Reviewer aYqm's novelty concern, while addressed, might benefit from their explicit acknowledgment. The theoretical contributions, though strengthened, remain relatively lightweight compared to the empirical work—a deeper statistical learning analysis could further solidify the theoretical foundation, though this may be beyond reasonable rebuttal scope.

**Reviewer Scores:**

Reviewer aYqm (Initial: 6): Likely would have maintained 6 or increased to 7. The rebuttal directly addressed the core novelty concern by clearly distinguishing PHAT's contributions from generic multi-frequency modeling, provided the requested mathematical formulation of periodic heterogeneity, and added two strong 2025 baselines (PatchMLP, TimeEmb) that PHAT outperforms. However, the reviewer's comment that "novelty is moderate" suggests they hold high standards for contribution claims. While the clarifications were thorough, without their explicit confirmation, a conservative estimate is they would have maintained 6 with increased confidence, with a reasonable possibility of moving to 7 given the comprehensive response.

Reviewer rCFS (Initial: 6): Would likely have increased to 7. All three specific weaknesses received direct, concrete responses: multi-level visualizations demonstrating negative components across data/feature/attention levels, FMPP ablation showing weighted fusion outperforms alternatives, and actual inference latency confirming efficiency. The reviewer's initial assessment was already positive ("simple and efficient method," "extensive experiments"), and the concerns were more about completeness than fundamental flaws. The thoroughness of the rebuttal addresses precisely what was requested.

Reviewer tCVf (Initial: 4 → 6): Would maintain 6. This reviewer already engaged during discussion and explicitly stated "I have increased my score" after receiving satisfactory responses on hyperparameter ranges, stress testing, and theoretical justification. The score increase from 4 to 6 represents substantial movement, and their concerns appear fully resolved. Further increase seems unlikely as they've already expressed satisfaction.

Reviewer DkU6 (Initial: 6): Would likely have increased to 7. The rebuttal provided exactly what was requested: comprehensive multi-level visualizations, expanded Figure 5 (three additional datasets) and Table 3 (full domain coverage), and in-depth discussion with ablations on bucket assignment and cross-bucket interaction strategies. The reviewer's initial assessment praised the motivation, clarity, and experimental setup; the concerns were about depth of analysis rather than fundamental issues. The substantial additions would likely merit a score increase.

---

### Decision · Program_Chairs · 2026-01-26

Accept (Poster)